# Inflammatory Modulation of Polyethylene Glycol-AuNP for Regulation of the Neural Differentiation Capacity of Mesenchymal Stem Cells

**DOI:** 10.3390/cells10112854

**Published:** 2021-10-22

**Authors:** Huey-Shan Hung, Wei-Chien Kao, Chiung-Chyi Shen, Kai-Bo Chang, Cheng-Ming Tang, Meng-Yin Yang, Yi-Chin Yang, Chun-An Yeh, Jia-Jhan Li, Hsien-Hsu Hsieh

**Affiliations:** 1Graduate Institute of Biomedical Science, China Medical University, Taichung 40402, Taiwan; doraemon_blue1201@hotmail.com (W.-C.K.); kbwork2021@gmail.com (K.-B.C.); fireleafmaple@hotmail.com (C.-A.Y.); betty79292@gmail.com (J.-J.L.); 2Translational Medicine Research, China Medical University Hospital, Taichung 40402, Taiwan; 3Department of Neurosurgery, Neurological Institute, Taichung Veterans General Hospital, Taichung 407204, Taiwan; shengeorge@yahoo.com (C.-C.S.); yangmy04@gmail.com (M.-Y.Y.); jean1007@gmail.com (Y.-C.Y.); 4Department of Physical Therapy, Hung Kuang University, Taichung 433304, Taiwan; 5Basic Medical Education Center, Central Taiwan University of Science and Technology, Taichung 406053, Taiwan; 6College of Oral Medicine, Chung Shan Medical University, Taichung 40201, Taiwan; ranger@csmu.edu.tw; 7Blood Bank, Taichung Veterans General Hospital, Taichung 407204, Taiwan; hhhsu@vghtc.gov.tw

**Keywords:** polyethylene glycol, gold nanoparticles, mesenchymal stem cells, differentiation, tissue regeneration

## Abstract

A nanocomposite composed of polyethylene glycol (PEG) incorporated with various concentrations (~17.4, ~43.5, ~174 ppm) of gold nanoparticles (Au) was created to investigate its biocompatibility and biological performance in vitro and in vivo. First, surface topography and chemical composition was determined through UV-visible spectroscopy (UV-Vis), Fourier-transform infrared spectroscopy (FTIR), atomic force microscopy (AFM), scanning electron microscopy (SEM), free radical scavenging ability, and water contact angle measurement. Additionally, the diameters of the PEG-Au nanocomposites were also evaluated through dynamic light scattering (DLS) assay. According to the results, PEG containing 43.5 ppm of Au demonstrated superior biocompatibility and biological properties for mesenchymal stem cells (MSCs), as well as superior osteogenic differentiation, adipocyte differentiation, and, particularly, neuronal differentiation. Indeed, PEG-Au 43.5 ppm induced better cell adhesion, proliferation and migration in MSCs. The higher expression of the SDF-1α/CXCR4 axis may be associated with MMPs activation and may have also promoted the differentiation capacity of MSCs. Moreover, it also prevented MSCs from apoptosis and inhibited macrophage and platelet activation, as well as reactive oxygen species (ROS) generation. Furthermore, the anti-inflammatory, biocompatibility, and endothelialization capacity of PEG-Au was measured in a rat model. After implanting the nanocomposites into rats subcutaneously for 4 weeks, PEG-Au 43.5 ppm was able to enhance the anti-immune response through inhibiting CD86 expression (M1 polarization), while also reducing leukocyte infiltration (CD45). Moreover, PEG-Au 43.5 ppm facilitated CD31 expression and anti-fibrosis ability. Above all, the PEG-Au nanocomposite was evidenced to strengthen the differentiation of MSCs into various cells, including fat, vessel, and bone tissue and, particularly, nerve cells. This research has elucidated that PEG combined with the appropriate amount of Au nanoparticles could become a potential biomaterial able to cooperate with MSCs for tissue regeneration engineering.

## 1. Introduction

Traumatic nerve injuries contribute to a high proportion of clinical symptoms and commonly cause the loss of motor/sensory functions, degeneration of nerve fibers, and even the death of neurons [1,2]. However, the self-regeneration of neurons is slow and usually incomplete. To overcome these barriers, creating an appropriate accessory material may promote efficiency in neuron regeneration [3]. A previous study had verified that neural stem cells (NSCs) could well restore neuronal functions due to their self-renewal and multipotential properties [4]. In spite of their advanced abilities, NSCs still require extra bioactive molecules to maintain cell survival and promote differentiation ability while also taking on the risk of developing into a tumor after being implanted at the area of injury [5].

Strategies to choose appropriate nano biomaterials and cell lines are necessary for tissue regeneration engineering [6]. Poor biocompatibility and lack of differentiation capacity may lead to failure in clinical treatments [7]. Thus, the mechanical and biological properties of nanomaterials such as biodegradation, cytotoxicity, and the ability of the anti-immune response should be of high concern [8]. Poly(e-caprolactone) (PCL) is the most commonly used semi-crystalline polyester in tissue repair owing to its hydrophobic and low biodegradation properties [9]. However, after PCL was synthesized with an ester-based polyurethane (PU), the hydrophilicity and biodegradation rates were improved [10]. Furthermore, polyethylene glycol (PEG) was shown to be both highly hydrophilic and biocompatible regarding the synthesis of hydrogels in drug delivery and tissue regeneration treatment. Published literature has demonstrated that PEG-PU hydrogels presented crystallinity involving PEG segments. The evidence of cytotoxicity assay also indicated that PEG-PU hydrogels could promote cell proliferation [11]. A study revealed that combining cellulose acetate butyrate (CAB) nanofibers with hydrophilic polyethylene glycol (PEG) could improve the properties of CAB. CAB has limitations regarding tissue engineering due to its intrinsic hydrophobicity that causes a barrier for cell adhesion. After fabricating CAB with PEG, the CAB/PEG nanofibers demonstrated better hydrophilicity, with the results of cell viability in normal human dermal fibroblasts (NHDF) also indicating non-toxicity. The investigation of cell attachment elucidated that CAB/PEG nanofibers enhanced cell attachment ability more than pure CAB [12]. The above findings support PEG as being a potential biomaterial for tissue engineering through its ability to improve the properties of nanocomposites.

Previous literature has indicated that stem cell homing, migration, and differentiation properties are crucial for tissue repair/regeneration [13]. Regarding their superior self-renewal and easy-to-isolate abilities, as well as their less adverse foreign body reactions, mesenchymal stem cells (MSCs) are considered to be a promising therapeutic source. Indeed, MSCs have exhibited the ability to perform anti-immune responses [14]. Additionally, MSCs secrete various types of growth factors, cytokines, and ECM-degrading proteases to facilitate tissue regeneration [15]. Stromal cell-derived factor α (SDF-1α) is a prominent chemokine which enhances cell migration [16], with the interaction between SDF-1α and C-X-C chemokine receptor type 4 (CXCR4) acting dominant in the efficiency of migration [17,18]. A study also investigated the mechanism of SDF-1α/CXCR4 signaling involved in MSC migration after transplanting MSCs into brain injury rats. The results demonstrated SDF-1α was highly expressed in the injured area to stimulate MSC migration. Furthermore, the authors used BrdU to identify the newly generating cells and double label GFAP for astrogliosis. The amount of BrdU/GFAP-positive cells were also increased in the injured area. The above evidence supported SDF-1α as a critical factor for cell migration [19]. Previous research has shown that MSCs could inhibit pro-inflammatory responses and reduce oxidative stress in in vivo models. MSCs promoted tissue regeneration processes through constructing an appropriate microenvironment to recruit stem cell-like progenitor cells, which are then differentiated into various types of cells, including bone, endothelial and even fat cells [20,21]. Furthermore, other literature has also reported that MSCs could be induced to process neuronal differentiation. The MSCs were observed to become neuron-like cells with axonal and dendritic-like morphology, with the neurogenic differentiation markers nestin, Map2, Nefl, Tau, Flk and β-tubulin also investigated as being significantly expressed after MSCs were treated with β-mercaptoethanol [22]. Therefore, MSCs have become a potential tool for cell therapy and tissue regeneration engineering.

Nano-scale biomaterials such as metals [23], hydrogels [24], and graphene oxide [25] have been applied to various fields. These nano biomaterials have demonstrated efficiency in drug delivery [26], tumor therapy [27] and tissue regeneration [28]. In respect to gold nanoparticles (denoted as Au in the current study), owing to their advanced biocompatibility and controlled properties, they have been extensively studied and applied to various clinical applications such as cancer therapy, drug delivery, and regenerative medicine [29]. The Au nanoparticles can be easily synthesized and controlled at different sizes and nanostructures by surface plasmon resonance (SPR) [30]. Indeed, Au nanoparticles have demonstrated interactions with MSCs. For instance, in line with our previous research, after combining Au nanoparticles with a natural molecule, collagen, the results promoted MSC proliferation and attenuated immune responses. It also stimulated the expression of αVβ3 integrin/CXCR4, focal adhesion kinase (FAK), matrix metalloproteinase-2 (MMP-2), and Akt/endothelial nitric oxide synthase (eNOS) proteins in MSCs for angiogenesis and endothelialization [31]. Furthermore, a published study also investigated the effects of Au nanoparticles in myogenic differentiation, with the evidence demonstrating that Au nanoparticles facilitated the myogenic differentiation of myoblasts through upregulating the expression of myosin heavy chain (MHC) proteins and myogenic genes (MyoD, MyoG and Tnnt-1). In addition, Au nanoparticles also activated the p38α mitogen-activated protein kinase pathway (p38α MAPK) and promoted myogenic differentiation in skeletal muscle regeneration [32]. Moreover, a piece of previous literature also demonstrated the effect of Au nanoparticles in inducing mouse embryonic stem cells (ESCs) to differentiate into dopaminergic (DA) neurons. The results of that study indicated that Au nanoparticles could affect DA neuron differentiation through the mTOR/p70S6K signaling pathway [33]. The above evidence supports that Au nanoparticles should become a potential candidate for current research.

Furthermore, PEG was also applied in neuronal regenerative applications. A previous study created hydrogels containing gelatin methacrylate (GelMA) and polyethylene glycol (PEG) to culture with embryonic stem (ES) cells, and then investigated neuronal differentiation properties. The ES cells were observed to be highly viable, while also effectively differentiating into neuronal cells in the GelMA and PEG hydrogels. The study suggested that the GelMA hydrogel-encapsulated PEG microwell array could become a promising biomaterial for neuron disease treatments [34]. Another study also determined that the poly (ethylene glycol)—poly (L-alanine) aqueous solution (PEG-L-PA) facilitated neuronal-specific markers to be expressed in tonsil-derived mesenchymal stem cells (TMSCs). The evidence demonstrated that TMSCs showed multipolar elongation during PEG-L-PA 3D culturing. Indeed, the neuronal markers, including nuclear receptor-related protein (Nurr-1), neuron-specific enolase, microtubule-associated protein-2, neurofilament-M, and glial fibrillary acidic protein were all confirmed to be highly expressed in the PEG-L-PA culturing system. This evidence suggested that PEG combined with various materials may become an ideal model for neuronal regenerative medicine [35].

In this study, we have modified functionally AuNP-polyethylene nanocomposites (PEG-Au) and further explored the performance of PEG-Au nanocomposites on the immune modulation effect in MSC-based biomaterial applications. We have also assessed the potent differentiation capacity of PEG-Au nanocomposites for tissue regeneration.

## 2. Materials and Methods

### 2.1. Material Preparation

#### 2.1.1. Preparation of Polyethylene Glycol (PEG)

Polyethylene glycol (PEG) was purchased from Sigma-Aldrich, Burlington, MA, USA (500 mM, with an average molecular weight = 200 kDa). The PEG solution was diluted 25 times with ddH_2_O to achieve a final concentration of 20 μM (1 mL of PEG and 24 mL of ddH_2_O). The solution was then cast to culture dishes and 6-, 24-, and 96-well plates for 30 min to sufficiently contact with the surface of the culture plates. The excess solution was removed to obtain a nano thin film and was subsequently applied to the following experiments.

#### 2.1.2. Preparation of Polyethylene Glycol-Gold Nanoparticles (PEG-Au)

The gold nanoparticle (AuNPs) solutions used in the current study were purchased from Gold Nanotech Inc. (Taipei, Taiwan). The AuNPs were collected according to a previous study [36] and dispersed into distilled water (~50 ppm). The diameter of the AuNPs was approximately 3–5 nm. The polyethylene glycol-gold nanoparticle (denoted as PEG-Au) solutions were prepared by mixing PEG solution with various amounts of AuNPs (~17.4, ~43.5, and ~174 ppm), then sonicated for 30 min. Each solution was coated on the surface of culture dishes and culture plates for 30 min to form PEG-Au thin films. The residual solution was then removed to process for use in upcoming experiments.

#### 2.1.3. Reagents of Differentiation Assay

For neural differentiation, 1 mM final concentration of β-mercaptoethanol (1 mM, Sigma, USA) was used. For osteogenic differentiation, dexamethasone (0.1 μM, Sigma, USA) and ascorbic acid-2-phosphate (0.25 mM, Sigma-Aldrich, Burlington, MA, USA) were used. For adipogenic differentiation, dexamethasone (0.1 μM, Sigma, USA), human insulin (0.5 μM, Sigma, USA), and indomethacin (30 μM, Sigma, USA) were used.

### 2.2. Material Characterization

#### 2.2.1. UV-Visible Spectroscopy

The spectra of pure Au and PEG-Au (~17.4, ~43.5, ~174 ppm) were measured using a Helios Zeta spectrophotometer (ThermoFisher, Pittsfield, MA, USA), with the wave range being from 200 to 800 nm. A peak of 520 nm was the absorption wavelength of the Au nanoparticles. To measure the sample, a quartz colorimetric tube was first cleaned with deionized water and wiped dry with mirror paper before having deionized water added. It was then measured for background absorption. Afterwards, each sample was measured sequentially. To avoid any possible factors which would affect the precision of the results, the quartz tube had to be repeatedly cleaned with deionized water in order to remove any residuals from the previous sample. Origin Pro 8 (Originlab Corporation, Northampton, MA, USA) software was applied to measure and quantify the results.

#### 2.2.2. Fourier Transform Infrared Spectroscopy (FTIR) Analysis

Infrared (IR) spectra were measured by a Fourier transform IR spectrometer (Shimadzu Pretige-21, Kyoto, Japan). A 0.06 g solution of potassium bromide (KBr, Sigma-Aldrich, Burlington, MA, USA) was mixed with 200 mg of each sample of pure PEG and PEG-Au (~17.4, ~43.5, ~174 ppm). The mixtures were then independently scanned a total of eight times under the scanning range of 500–4000 cm^−1^ with a 2 cm^−1^ resolution to acquire the spectrum [37].

#### 2.2.3. Free Radical Scavenging Ability and Hydrophilicity Property

DPPH (2,2-diphenyl-1-picrylhydrazyl) (Sigma-Aldrich, Burlington, MA, USA) was used to investigate the free radical scavenging ability of pure PEG and PEG-Au (~17.4, ~43.5, ~174 ppm) nanocomposites in the current study [38]. One (1) ml of distilled water containing each sample was mixed with 3 mL DPPH in methanol and then allowed to interact for 90 min. The absorbance wavelength of each sample was measured at 539 nm by a UV-visible spectrophotometer (Helios Zeta, Thermo, Waltham, MA, USA). The results of the free radical scavenging ability were calculated according to the formula: scavenging ratio (%) = [1 − (absorbance of test sample/absorbance of control)] × 100%.

The hydrophilicity property of the pure PEG and PEG-Au composites was also investigated. Each sample was loaded on silicon substrates with 0.7 μL of distilled water added dropwise on the surface to measure the water contact angles. The experiment proceeded through a PGX model instrument at RT for quadruplicates.

#### 2.2.4. Atomic Force Microscopy (AFM)

The surface morphology of the pure PEG and PEG-Au composites was investigated by an MFP-3D atomic force microscope (Asylum Research, Santa Barbara, CA, USA). Firstly, 100 μL of each sample was cast in a silicon wafer and then allowed to air dry. Next, a silicon cantilever (Olympus AC240TS) with low-noise characteristics was applied to observe the topography maps, with the range of the spring constant approximately 2.0 N/m. Topography images were then captured in AC mode at 512 × 512 pixels, with the average roughness analyzed via Image J 5.0 software (Media Cybernetics Inc Rockville, MD, USA).

#### 2.2.5. Transmission Electron Microscope (TEM)

TEM images were acquired from a JEM 1010 electron microscope (JEOL Ltd., Akishima, Tokyo, Japan) through setting the voltage at 80 keV in order to observe the particle size and structure. The sample was prepared by adding 5 μL of nanoparticle suspension onto a copper-coated TEM grid, then dried out at room temperature for observation.

#### 2.2.6. Dynamic Light Scattering (DLS) Measurement

The hydrodynamic diameter was investigated by a dynamic light scattering (DLS) analyzer (Malvern Zetasizer Nano ZS90, Taiwan) and analyzed with the software provided by the manufacturer. The experiment was processed at 25 °C, operating at a 532 nm light source with a 90 degree scattering angle. 1 mL of each sample (pure PEG and PEG-Au composites) was added into a cuvette for further measurements. The experiment was performed in triplicate.

### 2.3. Cell Culture and Characterization of Wharton’s Jelly-Derived Mesenchymal Stem Cells

The MSCs used in the current research were kindly supported by Prof. Woei-Cherng Shyu, which were acquired from the Wharton’s jelly tissue of a human umbilical cord [39]. The cells were cultured in high glucose Dulbecco’s modified Eagle’s medium (H-DMEM, Invitrogen) supplemented with 10% FBS, 1% (*v*/*v*) antibiotics (100 U/mL penicillin/streptomycin) and 1% sodium pyruvate.

The specific surface antigens of the MSCs were characterized through flow cytometry [40]. The MSCs were harvested and detached with 2mM EDTA in phosphate-buffered saline (PBS) and washed with PBS containing 2% bovine serum albumin (BSA) and 0.1% sodium azide (Sigma-Aldrich, Burlington, MA, USA). Then, the MSCs were incubated with antibodies conjugated with fluorescein isothiocyanate (FITC), phycoerythrin (PE) or PerCP-Cy5.5 against the indicated markers: CD14-FITC, CD34-FITC, CD45-FITC, CD44-PE, CD90-PerCP-Cy5.5, and CD105-FITC (BD Pharmingen, San Diego, CA, USA). Further, PE-conjugated IgG1 and FITC-conjugated IgG1 (BD Pharmingen) were applied as isotype controls. Ultimately, the MSCs were analyzed by FACS software (Becton Dickinson LSR II, Canton, MA, USA). The cells at the 8^th^ passage were used in the current research.

### 2.4. Biocompatibility Assay

#### 2.4.1. Examination of Cell Viability

The MSCs were seeded at a density of 6 × 10^3^ cells per well in a 96-well plate coated with pure PEG and PEG-Au with various concentrations of Au nanoparticles (~17.4, ~43.5, ~174 ppm). After incubating for 24, 48, and 72 h, the MTT [3-(4, 5-cimethylthiazol-2-yl)2, 5-diphenyltetrazolium bromide] solution (0.5 mg/mL) was added to each well for 3 h incubation at 37 °C. Later, dimethyl-sulfoxide (DMSO) solution was added and then incubated for 10 min to dissolve the crystals. The absorbance at 570 nm was read by an SpectraMax M2 reader (Molecular Devices, San Jose, CA, USA).

#### 2.4.2. Examination of Reactive Oxygen Species (ROS) Generation

The production of reactive oxygen species (ROS) was targeted by an oxidation-sensitive fluorescent probe, DCFH-dA (2, 7dichlorofluorescin diacetate) (Sigma-Aldrich, Burlington, MA, USA). MSCs at a cell density of 2 × 10^5^ were added onto 6-well plates coated with pure PEG or PEG-Au nanocomposites (~17.4, ~43.5, ~174 ppm) and then incubated at 37 °C for 48 h. After the 48 h, the cells were cautiously washed twice with phosphate-buffered saline (PBS), then stained with 10 nM DCFH-dA in the dark at 37 °C for 30 min. The intracellular ROS was detected through use of a flow cytometer (Becton Dickinson, Canton, MA, USA) and was quantified by using Flow Jo 7.6 (Becton Dickinson, Canton, MA, USA) software [31].

#### 2.4.3. Monocyte and Platelet Activation Test

Human monocytes were collected from the whole blood of healthy volunteers following Percoll protocol (Sigma-Aldrich, Burlington, MA, USA) [41] with IRB approval (CE12164) from the Taichung Veteran Hospital. The monocytes (1 × 10^5^/well) were seeded on a 24-well plate coated with different materials and incubated in an RPMI medium conditioned medium [10% FBS and 1% (*v*/*v*) antibiotics (10,000 U/mL penicillin G and 10 mg/mL streptomycin)] for 96 h at 37 °C. Afterwards, the cells were separated on a 24-well plate using 0.05% trypsin. The ratio of monocytes and macrophages was observed under a microscope. To further confirm the inflammatory response, CD68 (as a marker of macrophages) was also investigated through primary anti-CD68 antibody (GeneTex Inc, Irvine, CA, USA) immunofluorescence staining.

Platelets (2 × 10^6^/well) were cultured on various materials for 24 h of incubation, then fixed with a 2.5% glutaraldehyde solution for 8 h. After incubating for 8 more hours, the platelets were washed twice with PBS and then dehydrated by adding a 30% to 100% concentration of alcohol after standing at room temperature for 10 min during each step. After being dried at a critical point, the morphology of the platelets cultured with different materials was observed using SEM (JEOL JEM-5200, JEOL Ltd., Akishima, Tokyo, Japan).

#### 2.4.4. Cell Morphology and Adhesion Ability

Scanning electron microscopy (JEOL JEM-5200, JEOL Ltd., Akishima, Tokyo, Japan) was used to investigate the cell morphology and cell attachment ability while culturing with various materials. The MSCs (1 × 10^4^/mL) were cultured for 48 h and fixed with a 2.5% glutaraldehyde solution for 8 h. The cells were then dehydrated at a concentration of 30% to 100%. Finally, after being dried at a critical point, the cell morphology was investigated by SEM.

### 2.5. Biological Functional Examination

#### 2.5.1. Cell Migration Assay

The procedure of the cell migration assay was conducted according to a procedure reported in previous research [31]. The Oris seeding stoppers were cultured with 1 × 10^4^ MSCs per well and incubated for 24 and 48 h to reach confluency. Afterwards, the stoppers were removed, with the exception of one which was used in reference for pre-migration. The seeded plates were incubated at 37 °C for further investigation of pre-migration (t = 0 h) and post-migration (24 to 48 h). Afterwards, 200 μL of Calcein AM (2 μM, Sigma-Aldrich, Burlington, MA, USA) was added to each well and stained for 30 min at each time point, with the migration ability observed using a Zeiss Axio Imager A1 fluorescence microscope (White Plains, NY, USA). The semi-quantification of cell migration distance was acquired through Image J 5.0 software (Becton Dickinson, Canton, MA, USA).

#### 2.5.2. Immunofluorescence Staining

Firstly, 15 mm coverslip glasses were pre-coated with various materials and placed in a 24-well plate. MSCs at the density of 2 × 10^4^ cells per well were then seeded on these coverslip glasses and incubated in a conditioned medium. The cells were incubated with a 1:300 dilution of several primary antibodies, including CXCR4, nestin, GFAP and β-tubulin (Santa Cruz, TX, USA). Afterwards, the coverslip glasses were further washed and incubated with FITC-conjugated or PE-conjugated secondary antibodies (1:300 dilution) (Santa Cruz, TX, USA) for an additional 1 h. A 1 μg/mL DAPI solution (Invitrogen, White Plains, NY, USA) was then applied to locate cell nuclei. The samples were cautiously washed and placed on slides using a 50% glycerol/PBS solution before being sealed for subsequent observation. The images were captured in a darkroom using a fluorescence microscope (ZEISS AXIO IMAGER A1, White Plains, NY, USA). The fluorescent-positive cells were detected and quantified through Image J 5.0 software (Media Cybernetics, Burlington, MA, USA).

#### 2.5.3. Metalloproteinase Zymography Analysis

Cells (2 × 10^5^ per well) were seeded onto a 6-well plate overnight to reach attachment. The cultured medium was then collected after 48 h of incubation. A 10% SDS-PAGE gel containing 2% gelatin was used to separate the samples before the gel was then incubated with a denaturing buffer (pH 8.5, 40 mM Tris-HCl, 0.2 M NaCl, 10 mM CaCl2, and 2.5% Triton X-100) for 30 min at room temperature (RT). The gel was slowly stirred at room temperature and equilibrated with a development buffer (pH 8.5, 40 mM Tris-HCl, 0.2 M NaCl, 10 mM CaCl2, and 0.01% NaN3) for 18 h to reach activation in a 37 °C water bath. Next, the gel was stained with 0.2% Coomassie Brilliant Blue (10% acetic acid and 50% methanol) and washed with a destaining buffer (10% acetic acid, 20% methanol). After Coomassie Brilliant Blue staining, the protease-digested area would exhibit itself as clear bands in a dark blue background. The gel was then digitized using a densitometer, with MMP gelatinase activity quantified through Image J version 5.0 software (Media Cybernetics, Burlington, MA, USA).

#### 2.5.4. Enzyme-Linked Immunosorbent Assay (ELISA)

ELISA kits were acquired from R&D Systems (R&D, Minneapolis, MN, USA) to determine the SDF-1α, TNF-α, IL-1β, and IL-6 expression levels following the manufacturer’s instructions. In brief, cultured cells were enriched to obtain secretory proteins for future measurement. The protein samples were treated with primary antibodies for detection and secondary HRP-conjugated antibodies for signal amplification. The results were then analyzed through an enzyme-linked immunosorbent assay (ELISA) reader (SpectraMax M2, Molecular Devices, San Jose, CA, USA). The data were collected from 4 experiments.

#### 2.5.5. Western Blotting Experiment

The procedure for the Western immunoblotting assay was followed in the same manner as a previous study [42]. The collected cells were washed using PBS, and then, using a cell lysis buffer, were incubated for 60 min at 4 °C. The supernatant of cell lysates was collected by centrifuge for subsequent analysis. The protein samples were separated on SDS-PAGE before being transferred to a nitrocellulose membrane under optimal conditions for differentiation of Bcl-2, Bax, Cyclin D1, p21, and Caspase-3 using a 1:1000 dilution ratio of primary antibodies (Santa Cruz, TX, USA) and horseradish peroxidase (HRP)-conjugated secondary antibodies (Santa Cruz, TX, USA).

#### 2.5.6. Cell Cycle and Apoptosis Examination

Propidium iodide (PI) was used to stain the nuclei of the cells (2 × 10^5^) before cell cycle progress was investigated by a flow cytometer. Apoptotic cells were then investigated by following annexin V and PI double staining (Sigma-Aldrich, Burlington, MA, USA) protocols and quantified through a flow cytometer. Indeed, the cells that undergo apoptosis at an early stage tend to expose phosphatidylserine (PS) on the extracellular face of the plasma membrane, which can be specifically targeted by annexin V. The late stage of the apoptotic cells can be permeabilized by PI to target chromosomes. The images were acquired through a fluorescence microscope and semi-quantified via Image J 5.0 software (Media Cybernetics, Burlington, MA, USA).

### 2.6. Alizarin Red S (ARS) Staining

Intracellular calcium mineralization can be observed through Alizarin Red S (ARS) staining (Sigma-Aldrich, Burlington, MA, USA). Cells at the density of 1 × 10^5^ were seeded onto a 10 cm culture dish after incubating for 7 and 14 days. Afterwards, the cells were fixed with 4% PFA for 15 min and washed with PBS. A 2% ARS staining solution was then prepared and filtered while the pH was adjusted to 4.1~4.3. The cells were immersed in 500 μL of ddH2O for 1 min before being drained and removed. Afterwards, 500 μL of Alizarin Red S working solution was added after being reacted for 15 min at RT. Finally, the cells were drained and then soaked in 500 μL of deionized water for 2 min. The staining results were observed under a microscope (ZEISS AXIO IMAGER A1, White Plains, NY, USA).

### 2.7. Oil Red O (ORO) Staining

ORO histochemical analysis was applied to observe adipocyte differentiation. The MSCs at a density of 1 × 10^5^ were seeded on slides after incubation with various nanomaterials for 7 and 14 days. The MSCs were then fixed by 4% paraformaldehyde for 20 min and rinsed with 60% isopropanol before proceeding with the staining procedure using ORO (Sigma-Aldrich, 0.35% in isopropanol) and hematoxylin for 10 min. MSCs were then washed for two times with deionized water and dried at room temperature. The images of stained histology were captured by fluorescence microscopy for further semi-quantification of ORO-positive adipocytes.

### 2.8. Real-Time PCR Assay

RNA expression in the cells was extracted by Trizo1 (lnvitrogen, ThermoFisher, Waltham, MA, USA) and procedures were followed according to the manufacturer. The cells (1 × 10^5^ per well) were seeded in a 10 cm culture dish after incubating for 3, 5, and 7 days, then treated with 1 mL of Trizol for 5 min. Afterwards, the RNA was extracted by adding 200 μL of chloroform (Sigma, USA) for 15 s, then kept for 3 min at RT before being centrifuged at 12,000 rpm at 4 °C for 15 min. We then removed the supernatant and added 500 μL of isopropanol at 4 °C for 10 min of incubation. Finally, the samples were centrifuged at 12,000 rpm at 4 °C for 15 min. The supernatant was removed and washed twice with 1 mL of alcohol (75%). After drying the RNA, 20 μL of DEPC-treated H_2_O-soluble precipitate was added and quantified via reading the absorbance at 260 nm using a SpectraMax M2 ELISA reader (Molecular Devices, USA). A RevertAid First Strand cDNA DNA Synthesis Kit (Fermentas, Canada) was used to process cDNA synthesis. In short, 2 μL of oligo (dT) 18 primer and random hexamers (1:1) were added to the RNA sample and put into a gradient polymerase reaction temperature controller at 65 °C for 5 min. Afterwards, the addition of 4 μL of 5× reaction buffer, 1 μL of Lock RNase inhibitor (20 U/mL), 2 μL of dNTP Mix (10 mM) and 1 μL of RevertAid M-MuLV Reverse Transcriptase (200 U/mL) proceeded before being reacted at 42 °C for 60 min. Finally, each sample was carried out at 70 °C for 5 min to obtain cDNA. The polymerase chain reaction was carried out using the cDNA as a template and a 1Q2 Fast qPCR System with a reaction volume of 10 μL according to the manufacturer’s procedures. Firstly, 0.5 μL of primer (0.3 μM) and 5 μL of enzyme were added to the cDNA sample, with the RNA expression then analyzed using the Step One Plus Real-Time PCR System.

### 2.9. Rat Subcutaneous Implantation

In the current study, female Sprague Dawley (SD) rats (2–3 months old, 300–350 g) were used with approval from the Animal Care and Use Committee (La-1071565). The dorsal skin was cautiously incised at 10 mm to implant the materials [pure PEG and PEG-Au (~17.4, ~43.5, ~174 ppm) nanocomposites)] after local anesthesia. After implantation for 1 month, the wound tissue was resected for investigation. The formation of fibrous capsules in six sites was detected through hematoxylin and eosin (H&E) staining, with the average encapsulated fibrotic tissues being quantified using commercial software. The collagen deposition in each sample tissue was measured through use of a Masson’s trichrome staining kit by following the manufacturer’s instructions (Sigma, USA). To investigate the activation of macrophages, 1:200 dilutions of mouse monoclonal anti-CD86 and anti-CD163 antibodies (Santa Cruz, USA) were prepared to characterize the M1 and M2 polarization. Furthermore, 3-Amino-9-Ethylcarbazole (AEC) (ScyTek Laboratories, Inc, Logan, UT, USA) was used as a chromogen for immunohistochemical staining in the detection of CD45 expression. Subsequently, 20 μL of AEC chromogen concentrate was added to each 1 mL of AEC substrate buffer, with the mixture then applied to the tissue section following the instructions given for the procedure. APC anti-mouse CD45 antibodies were applied to investigate leukocyte infiltration. A 1:500 dilution of donkey anti-mouse IgG secondary antibodies (AF488, Invitrogen, USA) was applied for signal amplification. For analysis of the fluorescence intensity, an Olympus ix71 fluorescence microscope (Tokyo, Japan) was used. Moreover, TUNEL assay was applied to detect apoptotic cells in the rat models. During the late stages of cell apoptosis, DNA fragmentations could be targeted by labeling 3′-hydroxyl termini. An In Situ Cell Death Detection Kit, AP (Roche Diagnostics, Indianapolis, IN, USA) was purchased to detect the apoptotic cells by following the protocol provided by the manufacturer. Cell nuclei were targeted by DAPI. The number of rats was 5 (*n* = 5). Results were represented as mean ± SD.

### 2.10. Statistical Analysis

In the current study, samples for each experiment (*n* = 3~6) were collected, with the results represented as mean ± standard deviation (SD). All the experiments were independently processed in triplicate to avoid uncertainty. Student’s *t*-test and single-factor analysis of variance (ANOVA) methods were applied to evaluate the statistical difference amongst the various groups. The Bonferroni method was chosen for post hoc analysis in ANOVA. A *p* value less than 0.05 was considered statistically significant.

## 3. Results

### 3.1. Characterization of PEG Incorporating with Au Nanoparticles

The procedure for PEG-Au nanocomposite preparation is illustrated in Figure 1A. The schematic diagram elucidates that the combination of PEG and Au nanoparticles may create a strong interaction between oxygen atoms and Au nanoparticles. Afterwards, the absorption wavelength of Au nanoparticles was measured by UV-Vis spectroscopy, where pure Au nanoparticles had a typical peak of 520 nm, and were also observed in PEG-Au containing different amount of Au (~17.4, ~43.5, ~174 ppm) (Figure 1B). The FTIR spectra indicated that the specific peaks of pure PEG were 2931 cm^−1^ (-CH_2_ vibration), 2868 cm^−1^ (-CH_3_) and 1105 cm^−1^ (OH vibration), and these peaks were also found in PEG-Au 17.4, PEG-Au 43.5, and PEG-Au 174 ppm groups. The evidence indicated Au nanoparticles were successfully incorporated with PEG (Figure 1C). 

Furthermore, the free radical scavenging ability of pure PEG and PEG-Au nanocomposites was also investigated and is shown in Figure 1D. Based on the quantification results, PEG containing ~17.4 (15.7%), ~43.5 (20.1%) and ~174 (20.9%) ppm of Au nanoparticles demonstrated superior capture ability than simply pure PEG (*p* < 0.01). Moreover, the hydrophilicity property of biomaterials is crucial for cell attachment to the extracellular matrix (ECM) by adhesion molecules. Indeed, the water contact angles of pure PEG, PEG-Au 17.4, PEG-Au 43.5, and PEG-Au 174 ppm were 28.8°, 25.0°, 20.6°, and 24.2°, respectively. The results indicate that the addition of Au nanoparticles to the surface of the PEG polymer causes it to become more hydrophilic, particularly at the concentration of 43.5 ppm of Au nanoparticles, causing the nanocomposite to enhance the adhesion ability of MSCs (Figure 1E).

In addition, the surface morphology of pure PEG and PEG combined with different amounts of Au nanoparticles was observed by AFM (Figure 2A). Based on the AFM images, pure PEG was homogenous and uniform. When Au nanoparticles were combined with pure PEG, the surface morphology became smaller and strip-like. The PEG-Au nanocomposites were also observed using a transmission electron microscope (TEM) (Figure 2B). The surface roughness of pure PEG, PEG-Au 17.4, PEG-Au 43.5, and PEG-Au 174 ppm was 16.8 nm, 26.5 nm, 33.4 nm, and 27.9 nm, respectively (Figure 2C). In addition, the size of pure PEG, PEG-Au 17.4, PEG-Au 43.5, and PEG-Au 174 ppm was investigated by DLS assay, and the diameters were 3.48 nm, 3.1 nm, 4.4 nm, and 3.8 nm, respectively (Figure 2D). Furthermore, the cell morphology of MSCs migrating onto various materials was observed by a scanning electron microscope (SEM) (Figure 2E). When the MSCs migrated onto PEG-Au biomaterials, they would produce lamellipodia and filopodia to change their morphology for cell movement. 

### 3.2. Biocompatibility Assessments of PEG-Au Culturing with MSCs

The phenotypes of MSCs used in this study were firstly characterized through detecting specific surface markers using flow cytometry. The negative markers, such as CD14, CD34 and CD45, were highly expressed in hematopoietic cells, endothelial cells and immune cells, respectively. The specific antigens CD44, CD90 and CD105 for MSCs were significantly detected (Appendix A). The quantitative results analyzed using FACS software indicated more than 93% of positive markers and less than 1.01% of negative markers (Appendix A). Then, the MSCs were used in the following experiments. For further study, MTT assay evaluated that cell viability was significantly increased in PEG-Au groups, particularly in the PEG-Au 43.5 ppm group after incubating for 48 and 72 h (OD_570_ = 1.4 and 2.1, *p* < 0.01), followed by the PEG-Au 17.4 group (OD_570_ = 1.3 and 1.9, *p* < 0.05), PEG-Au 174 group (OD_570_ = 1.3 and 1.9, *p* < 0.05), and pure PEG group (OD_570_ = 1.2 and 1.9, *p* < 0.05), when compared to the control group (Figure 3A). Afterwards, the intracellular ROS generation of MSCs incubated with different materials was investigated at 48 h. The semi-quantified results indicate that the average amount of ROS production of pure PEG (~0.98-fold), PEG-Au 17.4 (~0.82-fold, *p* < 0.05), and PEG-Au 174 (~0.62-fold, *p* < 0.05) in the cells was higher than that in the PEG-Au 43.5 group (~0.22-fold, *p* < 0.01) when compared to the control group (Figure 3B), demonstrating that PEG-Au 43.5 had the superior anti-ROS generation ability.

Indeed, platelets and monocytes will rapidly activate when inflammatory response occurs. The degree of platelet activation from various materials was observed by SEM and is shown in Figure 3C. In the PEG-Au 43.5 group, the platelets were mostly represented by their rounded morphology (non-activated form), while cell cultures in pure PEG, PEG-Au 17.4, and PEG-Au 174 ppm were subsequently exhibited as flattened (active form). The semi-quantified data for platelet activation was lowest in the PEG-Au 43.5 group (~0.17-fold, *p* < 0.01), followed by PEG-Au 174 (~0.58-fold, *p* < 0.01), PEG-Au 17.4 (~0.74-fold, *p* < 0.01), and pure PEG (~0.83-fold). Furthermore, the expression of CD68 (a marker of macrophages) was measured by immunofluorescence staining, with the semi-quantitative result based upon fluorescence intensity showing the lowest expression as being in the PEG-Au 43.5 group (~0.30-fold, *p* < 0.01), followed by PEG-Au 17.4 (~0.38-fold, *p* < 0.01) and PEG-Au 174 (~0.45-fold, *p* < 0.01), when compared to the control group (Figure 3D,E). The monocyte conversion yield was lowest in the PEG-Au 43.5 group (~0.3-fold, *p* < 0.01), followed by PEG-Au 174 (~0.4-fold, *p* < 0.01), PEG-Au 17.4 (~0.5-fold, *p* < 0.01), and pure PEG (~0.9-fold). In addition, the expression of pro-inflammatory cytokines (TNF-*α*, IL-1β, IL-6) was also examined by ELISA assay (Figure 3F). The semi-quantitative data indicates that the lowest expression in the PEG-Au 43.5 group was at 24 h (TNF-*α:* ~0.69-fold, IL-1β: ~0.6-fold, IL-6: ~0.64-fold). The above evidence demonstrates that PEG-Au 43.5 ppm promoted superior anti-inflammatory abilities towards maintaining better biocompatibility.

### 3.3. Assessment of Migration Ability of MSCs by PEG-Au

The SDF-1α/CXCR4 pathway, as well as metalloproteinases (MMPs), each play important roles in cell migration for tissue repair. Thus, the expression of CXCR4 was analyzed through immunofluorescence, while SDF-1α secreted from MSCs was measured by ELISA assay. The fluorescence images of CXCR4 are shown in Figure 4A. The expression amount quantified by the IF method demonstrated that the PEG-Au 43.5 group had the greatest expression (~2.0-fold, *p* < 0.01), while the results measured by the FACS method also indicated a similar trend (PEG-Au 43.5: ~1.5-fold, *p* < 0.01) (Figure 4B,C). The amount of SDF-1α secreted from MSCs in the PEG-Au 43.5 (~1.1-fold, *p* < 0.01) and PEG-Au 17.4 groups (~1.1-fold, *p* < 0.01) was greater than other groups when compared to the control group. Furthermore, the MMP activity induced by different materials was investigated (Figure 4D). The expression of MMP-2 from MSCs in the PEG-Au 43.5 group was greatest at both 24 and 48 h (24 h: ~1.2-fold, 48 h: ~1.4-fold, *p* < 0.01). A similar trend occurred in the expression of MMP-9 in the PEG-Au 43.5 group (24 h: ~1.23-fold, 48 h: ~1.4-fold, *p* < 0.01) (Figure 4E,F). The real-time images obtained through Calcein-AM staining evaluated cell migration after culturing in various materials at both 24 and 48 h (Figure 4G). The semi-quantification of migration distance indicated that the PEG-Au 43.5 group (24 h: ~45-fold, 48 h: ~52.6-fold, *p* < 0.01) was significantly higher than other groups at both 24 and 48 h, followed by PEG-Au 174 (24 h: ~37.6-fold, 48 h: ~45.4-fold) and PEG-Au 17.4 (24 h: ~34.3-fold, 48 h: ~41-fold, *p* < 0.01) when compared to the control group (Figure 4H). The above findings suggest that PEG-Au 43.5 could enhance migration ability for tissue regeneration.

### 3.4. Effects of PEG-Au on the Expression of Apoptotic Related Proteins in MSCs

Western blotting assay was applied to examine the expression of the apoptotic related proteins Bcl-2, Bax, Cyclin D1, p21, and caspase-3 (Figure 5A). The quantitative results demonstrate that the anti-apoptotic proteins Bcl-2 and Cyclin D1 were remarkably expressed by MSCs in the PEG-Au 43.5 group (Bcl-2: ~1.9-fold, *p* < 0.01; Cyclin D1: ~2.6-fold, *p* < 0.01) when compared to other groups. Simultaneously, the pro-apoptotic proteins Bax as well as activated caspase 3 and p21 were significantly inhibited in the PEG-Au 43.5 group (Bax: ~0.5-fold, *p* < 0.01; activated caspase 3: ~0.2-fold, *p* < 0.01; p21: ~0.5-fold, *p* < 0.01) (Figure 5B). Moreover, the influence of various materials on the cell cycle of MSCs was investigated by flow cytometry to confirm their impact on cell survival and apoptosis. As presented in Figure 5C, the population of MSCs at the sub-G1 phase (presented as apoptosis) was significantly lower in each group. On the contrary, the population of MSCs at the S phase was remarkably higher in each group. Additionally, the results from the annexin V/PI double staining assay also proved that the pure PEG and PEG-Au nanocomposites could remarkably reduce the ratio of apoptotic MSCs (Figure 5D). These results strongly indicate that PEG-Au nanomaterials could prevent MSCs from apoptosis.

### 3.5. Multi-Differentiation Capacity of MSCs by PEG-Au

The differentiation capacity of MSCs induced by various materials was then evaluated. The phenotypes of neurons and endothelia were observed at days 3, 5 and 7, while the phenotypes of adipocytes and osteoblasts were investigated at days 5, 7, 14 and 21. After culturing MSCs with various materials, PEG-Au 43.5 ppm induced the highest expression level of neural markers at day 7 as compared to the control, with the immunostaining images shown in Figure 6A. The semi-quantitative data based on fluorescence intensity demonstrated an increase of ~19-fold for nestin (Figure 6B), ~9.7-fold for GFAP (Figure 6C), and ~4.9-fold for β-tubulin (Figure 6D) at day 7, compared to other groups. Adipocyte differentiation was investigated through ORO staining (Figure 6E), with the results indicating that there were more adipogenic cells, particularly in the PEG-Au 43.5 group at day 14 (~4.4-fold, *p* < 0.01) (Figure 6G). For osteogenic induction, the mineral deposition measured by ARS staining was discovered to be much greater in the PEG-Au 43.5 group (Figure 6F). The semi-quantitative data shows that the mineral differentiation in the PEG-Au 43.5 group was greatest at day 7 (~2.8-fold, *p* < 0.01), but slightly decreased at day 14 (~2.5-fold, *p* < 0.01) (Figure 6H). In addition, the gene expression level of neuronal, adipogenic and osteogenic markers was then analyzed by real-time PCR (Appendix A). As shown in Appendix A, PEG-Au 43.5 ppm significantly enhanced the expression of nestin mRNA (~14.7-fold, *p* < 0.01), GFAP mRNA (~9.7-fold, *p* < 0.01), and β-tubulin mRNA (~14.9-fold, *p* < 0.01) for neurogenic differentiation at day 7 compared to the control group. Appendix A indicates that the gene expression level of PPAR mRNA for adipogenic differentiation was significantly increased in the PEG-Au 43.5 group at day 7 (PPAR mRNA: ~1.5-fold, *p* < 0.01). Appendix A evaluates the expression of Runx-2 mRNA for osteogenic differentiation and shows that it was greatest in the PEG-Au 43.5 group at day 7 (Runx-2 mRNA: ~8.8-fold, *p* < 0.05). Amongst the four differentiation routes, PEG-Au nanocomposites remarkably induced differentiation abilities and particularly facilitated the neurogenic differentiation capacity of MSCs.

### 3.6. In Vivo Assessment of Biocompatibility and Anti-Inflammatory Response by PEG-Au

After subcutaneous implantation of various materials into rat models for one month, both the biocompatibility and inflammatory response were further investigated. The fibrous capsule formation was measured through H&E staining, and the semi-quantitative results indicate that the lowest formation was seen in the PEG-Au 43.5 group (~0.4-fold, *p* < 0.01), followed by PEG-Au 17.4 (~0.6-fold, *p* < 0.01), PEG-Au 174 (~0.8-fold, *p* < 0.01), and pure PEG (~1.1-fold) (Figure 7A,D). The collagen deposition was examined through Masson’s trichrome staining, where the results were also the lowest in the PEG-Au 43.5 group (~0.5-fold, *p* < 0.01), followed by PEG-Au 17.4 (~0.64-fold, *p* < 0.01), PEG-Au 174 (~0.64-fold, *p* < 0.01), and pure PEG (~0.98-fold) (Figure 7B,E). Moreover, the infiltration of CD45 (a marker of leukocytes) was stained to investigate the foreign body reaction from the implantation. The intensity of CD45 was then semi-quantified, with the results demonstrating that the PEG-Au 43.5 group induced the lowest expression (~0.64-fold, *p* < 0.01), followed by PEG-Au 17.4 (~0.74-fold, *p* < 0.05) and PEG-Au 174 (~0.7-fold, *p* < 0.05) (Figure 7C,F).

Subsequently, the markers of macrophages, CD86 and CD163, were selected as M1 and M2 polarization markers in order to evaluate the inflammatory response after implantation. Based on the fluorescence intensity, the expression level was semi-quantified. The results demonstrate that CD86 expression was lowest in the PEG-Au 43.5 group (~0.42-fold, *p* < 0.01), followed by PEG-Au 17.4 (~0.6-fold, *p* < 0.01) and PEG-Au 174 (~0.7-fold, *p* < 0.01) (Figure 8A,D). The expression of CD163 was more prominent in the PEG-Au 43.5 group (~1.5-fold, *p* < 0.01), followed by PEG-Au 17.4 (~1.3-fold, *p* < 0.01), PEG-Au 174 (~1.2-fold, *p* < 0.01) and pure PEG (~1.1-fold, *p* < 0.05) (Figure 8B,E). In addition, the endothilialization marker CD31 was also investigated, with the results indicating that the expression level was higher in each group compared to the control group [pure PEG: ~1.2-fold (*p* < 0.01), PEG-Au 17.4: ~1.24-fold (*p* < 0.01), PEG-Au 43.5: ~1.4-fold (*p* < 0.01), PEG-Au 174: ~1.2-fold (*p* < 0.01)] (Figure 8C,F).

The present research has elucidated that PEG-Au 43.5 ppm nanocomposites facilitate better biocompatibility, biological performance, and superior differentiation capacity, particularly for neural cells. In vivo assays also evaluated PEG-Au 43.5 ppm and discovered that it exhibited an advanced ability towards anti-immune response, as well as greater bio-safety.

## 4. Discussion

Nerves have a complex structure. The elasticity and tensile strength of nerve trunks and their ability to resist traction deformation depend on the fascicular tissue, and the epineurium provides a protective cushion against compression [43]. Nerve injuries are clinically common, difficult to self-repair or regenerate and are classified into three categories: neurapraxia, axonotmesis, and neurotmesis. The neuron undergoes various degenerative processes after complete axonal transection before attempting to regenerate. A growth cone distally regenerates and attempts to connect with the degenerated distal fiber. However, the success rate of nerve repair has not improved [44]. Therefore, treatment with nanomaterials in neural regeneration become an important issue. However, materials without good biocompatibility mechanical properties may lead to treatment failure and severe inflammation [45]. Several studies have pointed out the fact that autograft implantation procedures have had difficulties in neuron repair due to insufficient donor nerves and injury of the donor area [46], and allografts usually cause failures owing to transplant rejection and serious inflammation [47]. Other biomaterials, such as silicone nerve conduits, which is a traditional clinical material, also have shortcomings due to their nonresorbable nature and the fact that they cause a decrease in axonal conduction; in addition, they need to be removed via other treatments [48]. Thus, in recent years, nanomaterials have become a new field to improve neural regeneration engineering.

Carbon nanotubes, an artificial nanomaterial, were considered as a potential material in neuroscience approaches. A study proved that carbon nanotube scaffolds could affect neuronal signaling, as well as facilitate synaptic plasticity and the formation of neuron networks [49]. Moreover, synthetic polymer-based scaffolds have been created as tunable materials for tissue repair. Due to degradation in kinetics and mechanical strength properties, synthetic polymers have attracted attention. For instance, polyethylene glycol (PEG) [50], polylactic acid (PLA), and polyglycolic acid (PGA) [51] have all been well-investigated for utilization as tissue regeneration scaffolds. Specifically, PEG has been widely applied as a biomaterial owing to its bio-inert nature and can be easily modified to develop various structures. PEG has been combined with different polymers to provide novel treatments for tissue regeneration, such as nerve [52], bone [53], and vasculature [54] regeneration. Literature has proven that PEG decorated with biological adhesion peptide sequences could facilitate cell attachment [55]. Regarding immunogenicity, there are several FDA-approved compounds generated by PEGylation technology which are considered non immunogenic. However, a study had identified that PEGylated proteins stimulated the formation of antibodies against PEG. The PEG antibodies may cause the failure of the clinical therapy [56,57].

Nanoparticles such as Au have attracted attention in neuroscience field, and it also has been applied in various clinical approaches, such as biosensing, drug delivery and tissue regeneration engineering [58]. Due to its physical and chemical properties, Au nanoparticles have demonstrated high stability, low cytotoxicity and surface functionalized with polymers [59]. Au nanoparticles can easily interact with cells owing to their small size, which is approximately 1 to 100 nm. Our previous study demonstrated that natural molecules, such as collagen and fibronectin, fabricated with an appropriate amount of Au nanoparticles exhibited superior biocompatibility and induced advanced cell behavior [37,60]. Moreover, Au nanoparticles have been proved to enhance the growth of neural cells. A previous study indicated that chitosan-Au nanoparticles grafted onto poly(D,L-lactide) nerve conduits could induce regeneration in rat sciatic nerves [61]. Another study also demonstrated that near-infrared irradiation of Au nanorods can increase the number of neurons [62]. However, cetyltrimethylammonium bromide (CTAB) was usually used for the preparation of Au nanorods [63]. According to the research, the chemical residual of CTAB after the particle synthesis could induce cytotoxicty and disturb surface hydration of nanoparticles [64]. The physical Au nanoparticles used in the current research were collected as described in a previous study [36], demonstrating low cytotoxicity in vivo and in vitro assessments.

SDF-1α/CXCR4 pathways and matrix metalloproteinase (MMP-2/9) activities regulated stem cell migration and differentiation capacity [65]. A study demonstrated that the effect of PEGylated hollow gold nanoparticles (HGNs) could significantly enhance stem cell migration. Furthermore, PEG-HGNs also induced osteogenic differentiation [66]. Other previous research indicated that PEG can improve the mechanical properties of gelatin hydrogel. The evidence demonstrated that gelatin-PEG hydrogel could facilitate MSC migration as well as the adhesion and proliferation of MSCs [67]. Additionally, a piece of literature has verified that PEG inhibited cell apoptosis via its interactions with mitochondria. These results indicate that PEG improved mitochondrial function and inhibited the release of cytochrome c and pro-apoptotic factors [68]. A previous study also indicated that heparin-functionalized PEG hydrogels supported MSC viability [69].

PEG has been proven to increase the expression of neuronal expression markers in the presence of basic fibroblast growth factor (bFGF) and epidermal growth factor (EGF), and has also been proven to enrich the cell population of neural precursor cells (NPCs) [70]. Several studies have also indicated that the PEG-based hydrogel system enhanced adipose tissue regeneration [71] and osteogenesis [72]. In addition, a study also concluded that the properties of culture surface modification could significantly influence MSC behavior. Matrigel, the major components of which are laminin, collagen, entactin and heparan sulfate proteoglycans, at a coating density of 50 μg/cm^2^, could increase the amount of MSC-derived neuron-like cells with the best morphological differentiation and could strengthen cell expansion compared to the control group of an unmodified polystyrene surface [73]. This indicated that an appropriate amount of scaffold substrate can enhance both cell behavior and differentiation capacity. A recent study has demonstrated that a novel biodegradable hydrogel, oxidized alginate-gelatin-laminin (ADA-GEL-LAM) hydrogel, exhibited the ability to promote neuronal differentiation and proliferation of embedded human-induced pluripotent stem cells (hiPSCs) [74]. The research went on to add laminin (matrix component) to oxidized alginate-gelatin hydrogels, which effectively facilitated cell adhesion, migration, and differentiation while also suggesting ADA-GEL-LAM as a potential system for neural cell induction.

In accordance with our findings based upon culturing Wharton’s jelly MSCs with PEG-Au, the in vitro and in vivo assessments of biocompatibility, biological performance and differentiation capacity strongly suggest that combining MSCs with an optimal concentration of Au nanoparticles, ~43.5 ppm, provides an excellent microenvironment. After implanting nanomaterials into injured tissue, the foreign body responses such as inflammation and M2 macrophage polarization were of high concern [75,76]. Above all, decorating 43.5 ppm of Au nanoparticles on the PEG polymer significantly enhanced the neurogenic differentiation ability. Although our research developed a novelty nanomaterial, PEG-Au, with better biocompatibility, biological function, and neuronal differentiation capacity in vitro and in vivo, PEG-Au needs more in vivo assessments through combinations with nerve catheter and MSCs to further verify its efficiency to provide potential strategies for clinical treatments in neuronal regeneration engineering.

## 5. Conclusions

In this research, the nanocomposite, PEG-Au, was efficiently created in various concentrations. As demonstrated through in vitro assays, since PEG-Au exhibited advanced biocompatibility and biological performance for MSCs, particularly in the concentration of 43.5 ppm, it could superiorly enhance cell survival, migration and differentiation abilities when involved with the expression of CXCR4/SDF-1α and matrix metalloproteinase (MMP-2/9) activation. Additionally, this nanocomposite could also induce the further expression of neural-related protein (nestin, GFAP, and *β*-tubulin) guiding for the purpose of neurogenic differentiation ability. Furthermore, MSCs cultured on this nanomaterial also demonstrated the capacity for osteogenic and adipocyte differentiation. Simultaneously, after implanting the PEG-Au nanomaterial into a rat model, the results further indicated that the nanocomposite could improve endothelialization capabilities, the anti-inflammatory response, anti-fibrosis abilities and collagen formation, particularly for PEG-Au 43.5 ppm. In short, the findings demonstrated that PEG incorporating an appropriate amount of Au nanoparticles may be a potential biomaterial for neuronal regeneration engineering.

## Figures and Tables

**Figure 1 cells-10-02854-f001:**
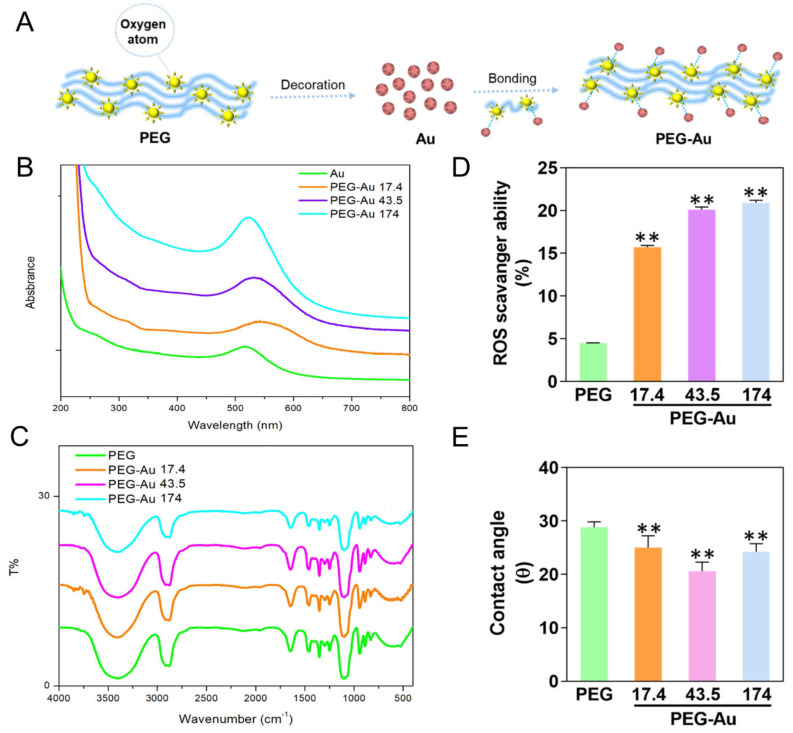
Schematic diagram illustration and material characterization. (**A**) The manufactured procedure of poly-ethyl glycol decorated with gold nanoparticle (PEG-Au). The strong interaction between Au nanoparticles and oxygen atoms contributed to the combination of PEG-Au nanocomposites. (**B**) The UV-Vis absorbance spectra. The peak at 520 nm indicates the absorbance of Au nanoparticles for pure Au and PEG combined with various concentrations of Au (~17.4, ~43.5, ~174 ppm). (**C**) The FTIR spectra of pure PEG and PEG-Au (~17.4, ~43.5, ~174 ppm). (**D**) The free radical scavenging ability of pure PEG and PEG in different concentrations of Au. The ROS scavenging ability of pure PEG was the lowest. On the contrary, after incorporating PEG with Au, the scavenging ability remarkably increased, particularly in the concentrations of ~43.5 and ~174 ppm. ** *p* < 0.01: greater than the control group. (**E**) The hydrophilicity property towards different biomaterials was also examined. The average water contact angle (θ) was semi-quantified, indicating that PEG-Au at 43.5 ppm was the lowest. The contact angle without water was θ = 0°. Data is represented as mean ± SD (*n* = 3). ** *p* < 0.01: less than the control group.

**Figure 2 cells-10-02854-f002:**
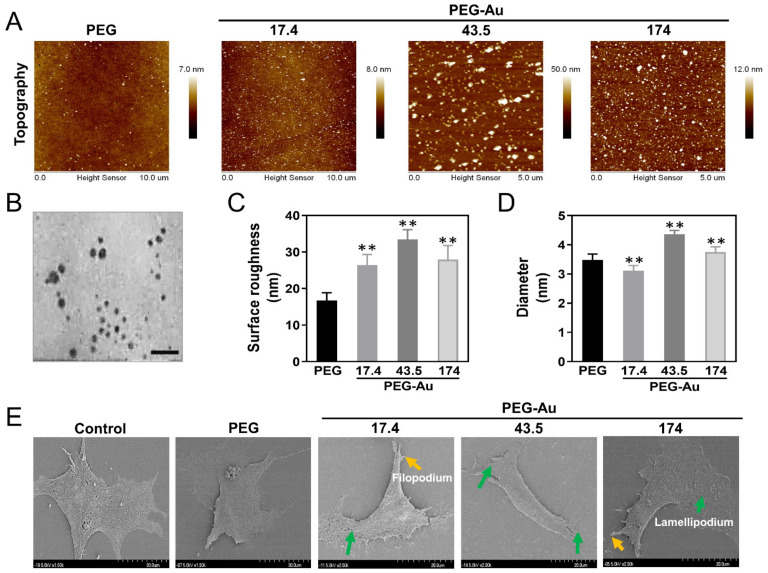
Characterization of the surface morphology and diameter property of pure PEG and PEG-Au nanocomposites. (**A**) The topography images for pure PEG and PEG-Au nanocomposites containing various concentration of Au (~17.4, ~43.5, ~174 ppm). (**B**) The TEM image of PEG-Au nanoparticles. (**C**) The value of root mean square (RMS) demonstrated in Ra is represented for average surface roughness. The results show that PEG with ~43.5 ppm of Au has the highest value of surface roughness. (**D**) The diameter of each PEG composite was investigated by DLS assay. The data demonstrates that the size of PEG-Au at 43.5 ppm was remarkably greater than the others. (**E**) The cell morphology of MSCs after being treated with pure PEG and PEG-Au nanocomposites was observed by SEM. The yellow arrow indicates filopodium, while the green arrow demonstrates lamellipodium. Data is represented as one of three independent experiments. The results are represented as mean ± SD. ** *p* < 0.01: greater than the control group.

**Figure 3 cells-10-02854-f003:**
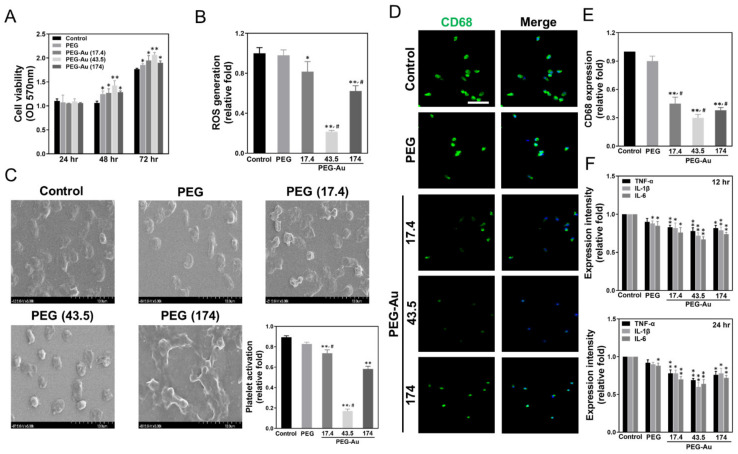
The biocompatibility performance of MSCs cultured with various biomaterials. (**A**) MSC proliferation investigated by MTT assay was promoted by culturing on pure PEG with different concentrations of Au nanoparticles (~17.4, ~43.5 and ~174 ppm) and then combining with PEG after various times periods of incubation (24, 48, 72 h). Data are expressed as mean ± SD (*n* = 3). * *p* < 0.05; ** *p* < 0.01: greater than the control group. All the results are representative of one of six independent experiments. (**B**) The intracellular ROS generation was semi-quantified by both 2,7-dichlorofluorescein diacetate (DCFH-dA) and flow cytometric analysis for MSCs incubated with different materials. Data are expressed as mean ± SD (*n* = 3). * *p* < 0.05; ** *p* < 0.01: smaller than the control group. All the results are representative of one of three independent experiments. (**C**) The SEM images demonstrate the adhesion and activation of human blood platelets in various materials. The data was semi-quantified by the degree of activation score. Data are expressed as mean ± SD (*n* = 3). ** *p* < 0.01: smaller than the control group. (**D**) The expression of CD68 for macrophages on different materials at 96 h. The cells were immunostained by primary anti-CD68 antibodies and conjugated with FITC-immunoglobin secondary antibodies (green color). DAPI was used to locate cell nuclei (blue color). Scale bar = 20 μm. (**E**) CD68 expression was then semi-quantified based upon fluorescence intensity. ** *p* < 0.01: smaller than the control group. (**F**) The expression of pro-inflammatory cytokines (TNF-α, IL-1β, IL-6) was measured at various times (8, 12, 24 h). The results were semi-quantified based upon fluorescence intensity. Data are expressed as mean ± SD (*n* = 3). * *p* < 0.05; ** *p* < 0.01: smaller than the control group. ^#^
*p* < 0.05: compared with PEG group.

**Figure 4 cells-10-02854-f004:**
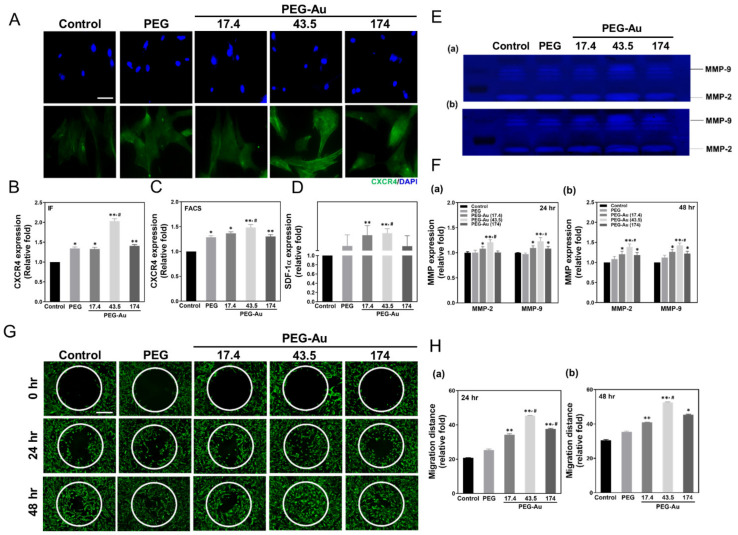
The expression of CXCR4/SDF-1α and MMPs, as well as the condition of cell migration for MSCs cultured with various materials. **(A**–**C**) The expression of CXCR4 in MSCs in various materials after 48 h of incubation. The cells were immunostained by primary anti-CXCR4 antibodies and then conjugated with FITC-immunoglobin secondary antibodies (green). DAPI was applied to locate cell nuclei (blue). Scale bar = 20 μm. The CXCR4 expression was then semi-quantified based upon fluorescence intensity by Image J software. Furthermore, the CXCR4 expression was also quantified by the FACS method. (**D**) The semi-quantified expression level of SDF-1α protein secreted from MSCs in various materials after 48 h of incubation. Data are expressed as mean ± SD (*n* = 3). (**E**,**F**) The MMPs enzymatic activity was measured by gelatin zymography assay at both (a) 24 and (b) 48 h. The semi-quantitative results are based upon the optical density (OD) of gelatinolytic bands through MMP expression. (**G**,**H**) The migration ability of MSCs cultured with various materials and observed at (a) 24 and (b) 48 h. MSCs migrating into the gap zone area were digitalized through fluorescence microscopy. The cells were stained by Calcein-AM (2 μM) prior to observation. Scale bar = 200 μm. The migration distance of MSCs was also semi-quantified. Data are expressed as mean ± SD (*n* = 3). * *p* < 0.05; ** *p* < 0.01: greater than the control group. ^#^
*p* < 0.05: compared with PEG group.

**Figure 5 cells-10-02854-f005:**
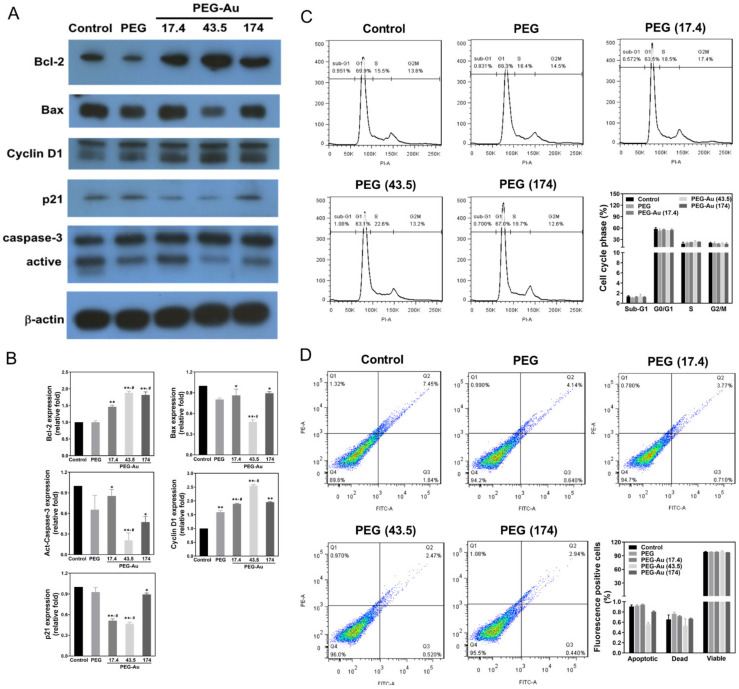
The expression of apoptosis-associated proteins and cell cycle progression of MSCs after culturing with various materials for 48 h of incubation. (**A**) Cell lysate was collected for Western blotting assay with primary anti-Bcl-2, anti-Bax, anti-cyclin D1, anti-p21, and anti-act-caspase-3 antibodies. β-actin was used as the control. (**B**) The expression of apoptosis-associated proteins was then semi-quantified. The results are represented as three independent experiments. Data are expressed as mean ± SD. * *p* < 0.05; ** *p* < 0.01 compared to the control group. (**C**) The cells were stained with propidium iodide (PI), and the DNA content of cells was examined by flow cytometry. Ten thousand (10,000) cells were counted in each sample, with the data computed by FACS software. (**D**) Apoptotic MSCs were also investigated by flow cytometry. The cells were co-stained with propidium iodide (PI) and annexin V-FITC. The quantification data of apoptotic and viable cells based upon the fluorescence positive cells is represented in percentages. The results are representative of three independent experiments. Data are presented as mean ± SD. ^#^
*p* < 0.05: compared with PEG group.

**Figure 6 cells-10-02854-f006:**
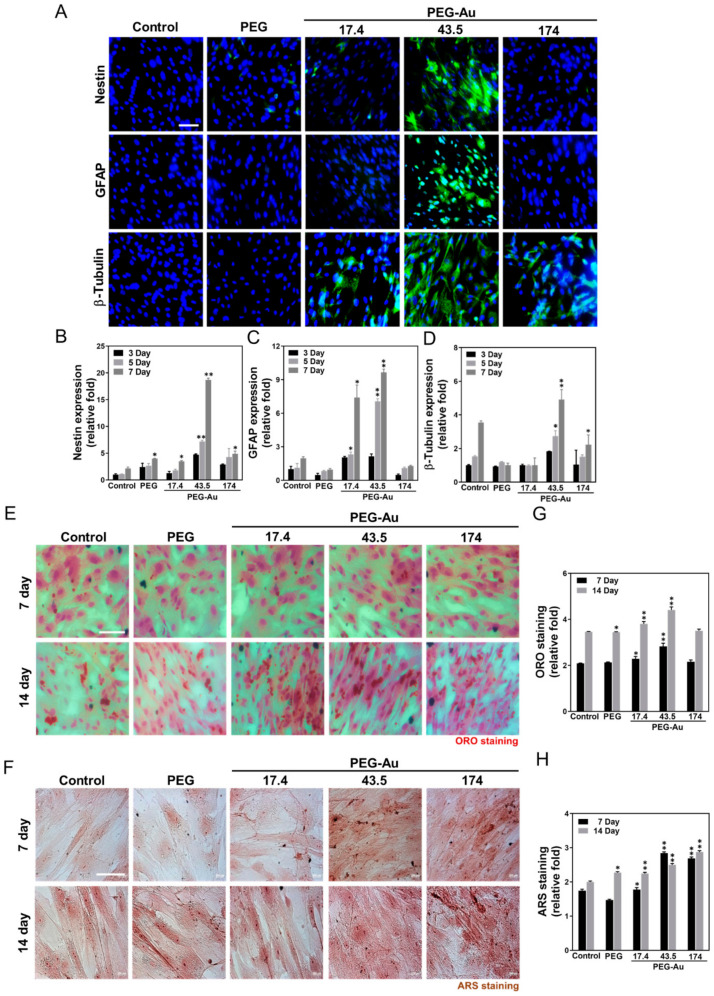
The neuronal differentiation ability of MSCs culturing in various materials. (**A**) The MSCs were stained with several primary neural-related markers, represented as nestin, GFAP, and β-tubulin antibodies, then conjugated with secondary FITC-immunoglobin secondary antibodies (green color). The cell nuclei were located using DAPI solution (blue color). Scale bar = 10 μm. The semiquantitative results of fluorescence intensity for (**B**) nestin (**C**) GFAP and (**D**) β-tubulin was measured by Image J software at various times (3, 5, 7 days), and indicate that under incubation with PEG-Au 43.5 ppm for MSCs, the expression of these three neural-related markers was the highest at day 7 when compared to other groups. Data are presented as mean ± SD. * *p* < 0.01; ** *p* < 0.01, greater than the control group. (**E**) Adipocyte differentiation of MSCs cultured with various materials was investigated using Oil Red O staining at day 7 and 14. (**F**) The osteogenic differentiation ability of MSCs was also investigated through use of Alizarin Red S (ARS) staining at day 7 and 14. (**G**) Quantification of adipocyte differentiation by ORO-positive lipid droplets of MSCs on different materials at days 7 and 14. (**H**) Semi-quantification of adipocyte differentiation was performed at days 7 and 14. Based upon the quantification results, the PEG-Au 43.5 ppm group enhanced adipocyte differentiation ability the greatest at day 14 when compared to other groups. Additionally, results from ARS staining indicate that PEG-Au at 43.5 ppm facilitated superior osteogenic differentiation ability at day 7, but this was slightly decreased at day 14. The results are representative of three independent experiments. Data are expressed as mean ± SD. * *p* < 0.05, ** *p* < 0.01, greater than the control group.

**Figure 7 cells-10-02854-f007:**
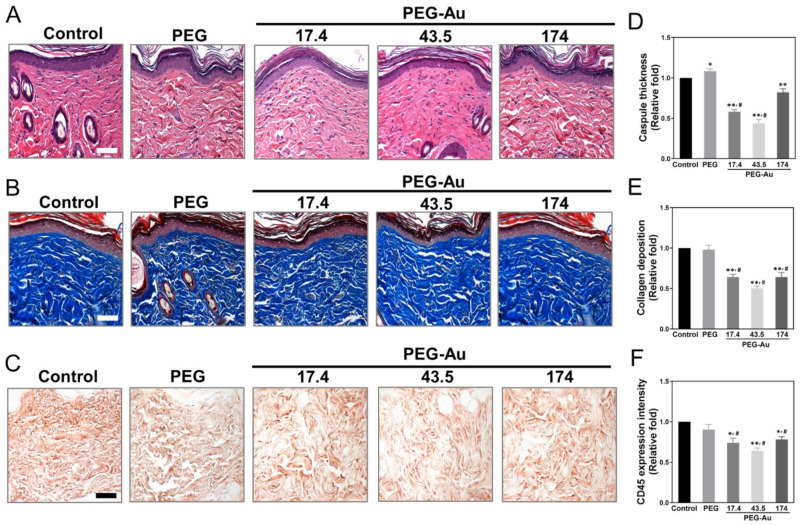
Evaluation of foreign body response after subcutaneous implantation of various materials for 4 weeks. (**A**) The capsule formation was performed using H&E staining. (**B**) The collagen deposition (blue color) was investigated through Masson’s trichrome staining. (**C**) The marker of immune inflammation, CD45, was stained in response to the implant materials using an APC kit. Scale bar = 100 μm. The quantification results are represented as (**D**) capsule thickness, (**E**) collagen deposition, and (**F**) CD45. The semi-quantified data show that PEG-Au nanocomposites could effectively decrease capsule formation, collagen deposition, and leukocyte infiltration in tissue, particularly in the PEG-Au 43.5 ppm group. The number of rats was 5 (*n* = 5). Data are expressed as mean ± SD. * *p* < 0.05, ** *p* < 0.01, smaller than the control group. ^#^
*p* < 0.05: compared with PEG group.

**Figure 8 cells-10-02854-f008:**
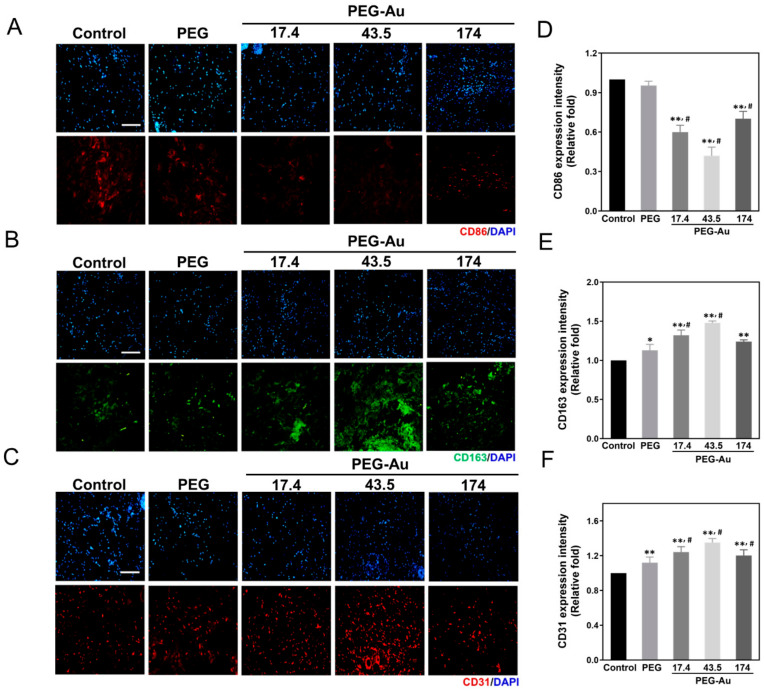
The images of immunohistochemical (IHC) staining in response to the implant materials. The markers of macrophages, (**A**) CD86 (M1, red color), (**B**) CD163 (M2, green color), and the endothelialization marker (**C**) CD31 (red color) were investigated. The fluorescence intensity of (**D**) CD86, (**E**) CD163, and (**F**) CD31 were also semi-quantified using Image J software. The quantified results indicate that PEG-Au 43.5 ppm significantly decreased the expression of CD86, but remarkably stimulated the expression of CD163. In addition, the endothelialization marker, CD31, displayed a higher expression in the PEG-Au group compared to the control group. The above findings elucidate that PEG-Au nanomaterials could attenuate macrophage activation, particularly in the PEG-Au 43.5 ppm group. Moreover, PEG-Au 43.5 could also induce the greatest expression of CD31 for endothelialization differentiation. Cell nuclei were stained by DAPI. Scale bar = 100 μm. Data are expressed as mean ± SD. (*n* = 5). * *p* < 0.05, ** *p* < 0.01: compared to the control group. ^#^
*p* < 0.05: compared with PEG group.

## Data Availability

Data are contained within the article.

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
