# Peer review of "Inflammatory Modulation of Polyethylene Glycol-AuNP for Regulation of the Neural Differentiation Capacity of Mesenchymal Stem Cells"

_cells, 2021, doi:10.3390/cells10112854_

Round 1

Reviewer 1 Report

The article is of interest and good quality. For this reason, it deserves to be published. Nevertheless, before accepting it, some errors need to be amended. First of all, those concerning the revision of your manuscript (e.g. line 111 there is a comment between authors). In the following, the scientific issues I recommend to improve:

  • Lines 72-73 the neurogenic markers you detailed are not of differentiation all of them. Please, revise.
  • Lines 83-84 not only the size of the AuNPs needs to be controlled. In fact, their shape is even more important. Please, add this information.
  • Section 2.3 à requires the inclusion of the ethical protocol approved to use human primary cells. Same for 2.4.3. this is really important and mandatory!
  • Revise legend of Figure 2 because it does not fit the figure.
  • There is a small controversy: while in the abstract you indicated CD45 as a labelling for neutrophiles, in the text (lines 623 and continuous), you stated this is a leukocytic marker. It is correct that it is a leukocytic marker, so the abstract must be changed.
  • Figure 9 is difficult to understand. Try to reformulate it.
  • In the discussion, I missed some comment regarding the immunogenicity attributed to some PEGs. I agree with you that you found nothing about that, but I think to be honest, a comment on that must be added.

Author Response

Reviewer 1:

The article is of interest and good quality. For this reason, it deserves to be published. Nevertheless, before accepting it, some errors need to be amended. First of all, those concerning the revision of your manuscript (e.g. line 111 there is a comment between authors). In the following, the scientific issues I recommend to improve:

  1. Lines 72-73 the neurogenic markers you detailed are not of differentiation all of them. Please, revise.

Answer:

Thanks the valuable comment from the reviewer. We have included the more detail description of neurogenic markers in the “Introduction” section “The MSCs were observed to become neuron-like cells with axonal and dendritic-like morphology, with the neurogenic differentiation markers, Nestin, Nef, Flk and β-tubulin also investigated as being significantly expressed after MSCs are treated with β-mercaptoethanol [22].” (Page 2-3, line 100-104)

  1. Lines 83-84 not only the size of the AuNPs needs to be controlled. In fact, their shape is even more important. Please, add this information.

Answer:

Thanks for the valuable suggestion from the Reviewer. We have included the more detail description in the “Introduction” section “The Au nanoparticles can be easily synthesized and controlled at different sizes. Indeed, gold nanostructure should include both scattering and absorption components by Surface Plasmon Resonance (SPR) [30].” (Page 3, line 112-113)

  1. Section 2.3 à requires the inclusion of the ethical protocol approved to use human primary cells. Same for 2.4.3. this is really important and mandatory!

Answer:

Thanks for the valuable suggestion from the Reviewer. We have included the description in the “Section 2.3” “The MSCs used in the current research were kindly supported by Prof. Woei-Cherng Shyu, which were acquired from the Wharton’s jelly tissue of a human umbilical cord [39].” and in Section 2.4.3 “Human monocytes were collected from the whole blood of healthy volunteers following Percoll protocol (Sigma-Aldrich, Burlington, MA, USA) [41] with IRB approval (CE12164).” (Page 5, line 238-239) and (Page 6, line 276-278)

  1. Revise legend of Figure 2 because it does not fit the figure.

Answer:

We have modified the legend of Figure 2 to fit the figure “Figure 2. Characterization of the surface morphology and diameter property of pure PEG and PEG-Au nanocomposites. (A) The topography images for pure PEG and PEG-Au nanocomposites containing various concentration of Au (~17.4, ~ 43.5, ~ 174 ppm). (B) The TEM image of PEG-Au nanoparticles. (C) The value of Root Mean Square (RMS) demonstrated in Ra is represented for average surface roughness. The results show that PEG with ~ 43.5 ppm of Au has the highest value of surface roughness. (D) The diameter of each PEG composite was investigated by DLS assay. The data demonstrates that the size of PEG-Au at 43.5 ppm was remarkably greater than the others. (E) The cell morphology of MSCs after being treated with pure PEG and PEG-Au nanocomposites was observed by SEM. The yellow arrow indicates filopodium, while the green arrow demonstrates lamellipodium. Data is represented as one of three independent experiments. The results are represented as mean ± SD. **p < 0.01: greater than the control group.” (Page 11-12, line 493-501)

  1. There is a small controversy: while in the abstract you indicated CD45 as a labelling for neutrophiles, in the text (lines 623 and continuous), you stated this is a leukocytic marker. It is correct that it is a leukocytic marker, so the abstract must be changed.

Answer:

Thanks for the valuable comment from the reviewer. We have changed the wording in the “Abstract section” section “neutrophil cells infiltration (CD45)” to “leukocytes infiltration (CD45)”. (Page 1, line 37); in the “Materials and Methods” “leukocytes infiltration” (Page 9, line 440); in the Figure caption section “leukocytes infiltration” (Page 19, line 700) 

  1. Figure 9 is difficult to understand. Try to reformulate it.

Answer:

We have changed Figure 9 into Graphical Abstract.

  1. In the discussion, I missed some comment regarding the immunogenicity attributed to some PEGs. I agree with you that you found nothing about that, but I think to be honest, a comment on that must be added.

Answer:

Thanks for the valuable suggestion from the Reviewer. We have added the description in the “Discussion” section “Regarding the immunogenicity, there are several FDA-approved compounds generated by PEGylation technology were considered non immunogenic. The immunogenicity of PEGylated therapeutic agents in clinical use. Additionally, PEG has been compared to hapten a nonimmunogenic compound able to trigger antibody induction when interfaced with an immunogenic protein [56, 57].” (Page 20, line 761-765)

Reviewer 2 Report

Dear authors
I found your idea very interesting but I have had great difficulty reading the paper. The first thing I would like to tell you concerns the research design. I have the impression that the desire to include all the experiments is at the expense of a fluid reading for the reader. Therefore, I recommend dividing the paper into two parts: the first relating to the properties of the compound and the second relating to the role of the compound in the differentiation of MSCs. 

Therefore I believe that the paper can be accepted only with this simplification to make it more fluid and less confusing.

Minor revision

- the section "material and methods" lack the characterization of the MSCs as well as the approval of the ethics committee.

- within the text, there are some comments in brackets (line 111, 222).

- it is absolutely necessary to divide the results section.

- too many pictures in the supplementary material. Could be better to use them to enrich the new papers

- the authors write that Peg-Au increases the proliferation of MSC but this is in contrast with the idea that Peg-AU Increase multilineage differentiation of MSC. How can you explain that?

- Figure 5CD is unconvincing. how many times have you repeated the experiment?

- in figure 6b the author shows that nestin increased 20 times in peg-Au 43.5 (7 days) and at the same time also GFAP and b3-tubulin increased. it is very strange because nestin is a neural stem cells marker and increases precociously and then diminished. How can you explain that?  besides, I don't understand how you stimulated neuronal, adipocytic, and osteogenic differentiation. Did you use only Peg-AU or other compounds?

Author Response

Reviewer 2:

Dear authors
I found your idea very interesting but I have had great difficulty reading the paper. The first thing I would like to tell you concerns the research design. I have the impression that the desire to include all the experiments is at the expense of a fluid reading for the reader.

  1. Therefore, I recommend dividing the paper into two parts: the first relating to the properties of the compound and the second relating to the role of the compound in the differentiation of MSCs. reviewer

Answer:

Thanks the value comment from reviewer. We have modified the manuscript accordingly by reviewer’s suggestion as list below.

  1. Therefore, I believe that the paper can be accepted only with this simplification to make it more fluid and less confusing.

Answer:

Thanks the valuable comment.

Minor revision

  1. the section "material and methods" lack the characterization of the MSCs as well as the approval of the ethics committee.

Answer:

We have include the surface marker characterization in the new Figure S1 and description in the “Materials and Methods” section “The MSCs used in the current research were kindly supported by Prof. Woei-Cherng Shyu, which were acquired from the Wharton’s jelly tissue of a human umbilical cord [39]. (Page 5, line 238-239); The specific surface antigen of the MSCs were characterized through a flow cytometry [40]. The MSCs were harvested and detached with 2mM EDTA in phosphate -buffered saline (PBS), and washed with PBS containg 2% bovine serum albumin (BSA) and 0.1 % sodium azide (Sigma-Aldrich, Burlington, MA, USA). Then the MSCs were incubated with antibodies conjugated with fluorescein isothiocyanate (FITC), phycoerythrin (PE) or PerCP-Cy5.5 against the indicated markers: CD14-FITC, CD34-FITC, CD45-FITC, CD44-PE, CD90-PerCP-Cy5.5, and CD105-FITC (BD Pharmingen, San Diego, CA, USA). Further, PE-conjugated IgG1 and FITC-conjugated IgG1 (BD Pharmingen) were applied as isotype controls. Ultimately, the MSCs were analyzed by FACS software (Becton Dickinson LSR II, Canton, MA, USA). The cells at the 8th passage were used in current research.” (Page 5, line 243-252); in the “Results” section “The phenotypes of MSCs used in this study were firstly characterized through detecting specific surface markers using flow cytometry. The negative markers such as CD14, CD34 and CD45, were highly expressed in hematopoietic cells, endothelial cells and immune cells, respectively. The specific antigen CD44, CD90 and CD105 for MSCs, were significantly detected (Figure S1A). The quantitative results analyzed using FACS software indicated more than 93% of positive markers and less than 1.01% of negative markers (Figure S1B). Then the MSCs were used in the following experiments.” (Page 12, line 518-524)

  1. Within the text, there are some comments in brackets (line 111, 222).

Answer:

We have removed the comments in brackets.

  1. It is absolutely necessary to divide the results section.

Answer:

Thanks the valuable comment by reviewer. We have removed some results (Figure 2B, Figure 9, Figure S1, Figure S2, Figure S3, Figure S4B, Figure S5) and make it to be more easy follow.

  1. Too many pictures in the supplementary material. Could be better to use them to enrich the new papers

Answer: 

We thank the valuable comment. We have removed the supplemental figure (Figure S1, Figure S2, Figure S3, Figure S4B, Figure S5) by reviewer’s suggestion. 

  1. The authors write that Peg-Au increases the proliferation of MSC but this is in contrast with the idea that Peg-AU increase multilineage differentiation of MSC. How can you explain that?

Answer:

We agreed the value comment from the reviewer. An idea substrate and/or scaffold for tissue engineering is stably to providing several critical properties of mechanical support, chemical stimuli, and biological signals for induction of cell adhesion and proliferation [ACS Nano 5, 4670-4678 (2011); Carbon 59, 200-211 (2013)]. However, the efficacy of PEG functionalized with gold nanoparticles on stem cell-derived tissue engineering is still unclear and need to be validated. In this study, we evaluated the effects of the as-fabricated AuNPs-decorated PEG nanocomposites for the biocompatibility and cell adherent efficiency on MSCs. This may attribute to the abounding surface chemical groups on the PEG and/or AuNPs deposition enhances the surface topography changes which are associated with cell adhesion and further affected cell proliferation. Indeed, surface features of nanocomposites including roughness, curvature, and wrinkled morphology are all affecting the cell adhesion and proliferation. Moreover, the nanotopographic network features on the surface of graphene nanogrid pattern were demonstrated that successful enhanced differentiation of MSCs. Besides, the induction the MSC had multilineage differentiation ability possibility due to addition different reagents. We have included the reagent in the “Materials and Methods” section “2.1.3. Reagents of Differentiation Assay For neural differentiation, 1 mM final concentration of β-mercaptoethanol (1 mM, Sigma, USA) was used. For osteogenic differentiation, dexamethasone (0.1 μM, Sigma, USA) and ascorbic acid-2-phosphate (0.25 mM, Sigma, USA) were used. For adipogenic differentiation, dexamethasone (0.1 μM, Sigma, USA), human insulin (0.5 μM, Sigma, USA), and indomethacin (30 μM, Sigma, USA) were used” (Page 4, line 171-176)

  1. Figure 5CD is unconvincing. how many times have you repeated the experiment?

Answer:

All results were obtained from three independent experiments and gained the same results. 

  1. In figure 6b the author shows that nestin increased 20 times in peg-Au 43.5 (7 days) and at the same time also GFAP and b3-tubulin increased. it is very strange because nestin is a neural stem cells marker and increases precociously and then diminished. How can you explain that?  besides, I don't understand how you stimulated neuronal, adipocytic, and osteogenic differentiation. Did you use only Peg-AU or other compounds?

Answer:

(1) We agreed the valuable comment from the reviewer. Nestin is a type VI intermediate filament (IF) protein. These intermediate filament proteins are expressed mostly in nerve cells where they are implicated in the radial growth of the axon. The intermediate filament (nanofilament) protein nestin is also expressed in neuroepithelial cells [1,2] and immature astrocytes in the developing central nervous system (CNS) [2,3], as well as in developing heart, muscle, kidney, and testis tissue [4-6]. In adult brain, nestin is expressed in NSPCs and in some astroglial cells (e.g., in the hippocampus) [7] and may be re-expressed in reactive astrocytes under pathophysiological conditions [8,9]. Mesenchymal stem cells is a kind of stem cells with multiple differentiation potential. Therefore, it will be necessary further assessment the differentiation ability of nestin expression conducted to biomaterials via surface fabricating with PEG-Au nanocomposites and implantation of MSCs into scaffold to evaluate the stem cells differentiation ability and evaluation the nerve regeneration capacity by vivo study.

Reference:

  1. Hockfield, S.; McKay, R. Identification of major cell classes in the developing mammalian nervous system. J Neurosci. 1985, 5, 3310-3328.
  2. Lendahl, U.; Zimmerman, L. B.; and McKay, R. D. CNS stem cells express a new class of intermediate filament protein. Cell 1990, 60, 585-595.
  3. Zerlin, M.; Levison, S. W.; Goldman, J. E. Early patterns of migration, morphogenesis, and intermediate filament expression of subventricular zone cells in the postnatal rat forebrain. J Neurosci. 1995, 15, 7238-7249.
  4. Kachinsky, A. M.; Dominov, J. A.; Miller, J. B. Intermediate filaments in cardiac myogenesis: nestin in the developing mouse heart. Journal of Histochemistry & Cutochemistry 1995, 43, 843-847.
  5. Fröjdman, K.; Pelliniemi, L.; Lendahl, U.; Virtanen, I.; and Eriksson, J. E. The intermediate filament protein nestin occurs transiently in differentiating testis of rat and mouse. Differentiation 1997, 61, 243-249.
  6. Chen, J.; Boyle, S.; Zhao, M.; Su, W.; Takahashi, K.; Davis, L.; DeCaestecker, M.; Takahashi, T.; Breyer, M. D. and Hao, C.-M. Differential expression of the intermediate filament protein nestin during renal development and its localization in adult podocytes. Journal of the American Society of Nephrology 2006, 17, 1283-1291.
  7. Filippov, V.; Kronenberg, G.; Pivneva, T.; Reuter, K.; Steiner, B.; Wang, L.-P.; Yamaguchi, M.; Kettenmann, H.; Kempermann, G. Subpopulation of nestin-expressing progenitor cells in the adult murine hippocampus shows electrophysiological and morphological characteristics of astrocytes. Molecular and Cellular Neuroscience 2003, 23, 373-382.
  8. Frisén, J.; Johansson, C. B.; Török, C.; Risling, M.; Lendahl, U. Rapid, widespread, and longlasting induction of nestin contributes to the generation of glial scar tissue after CNS injury. The journal of cell biology 1995, 131, 453-464.
  9. Eliasson, C.; Sahlgren, C.; Berthold, C.-H.; Stakeberg, J.; Celis, J. E.; Betsholtz, C.; Eriksson, J. E.; Pekny, M. Intermediate filament protein partnership in astrocytes. Journal of Biological Chemistry 1999, 274, 23996-24006.

(2) We have included the reagent in the “Materials and Methods” section “2.1.3. Reagents of Differentiation Assay For neural differentiation, 1 mM final concentration of β-mercaptoethanol (1 mM, Sigma, USA) was used. For osteogenic differentiation, dexamethasone (0.1 μM, Sigma, USA) and ascorbic acid-2-phosphate (0.25 mM, Sigma, USA) were used. For adipogenic differentiation, dexamethasone (0.1 μM, Sigma, USA), human insulin (0.5 μM, Sigma, USA), and indomethacin (30 μM, Sigma, USA) were used.” (Page 4, line 171-176)

Reviewer 3 Report

In the paper ‘Inflammatory Modulation of Polyethylene Glycol-AuNP for Regulation of the Neural Differentiation Capacity of Mesenchymal Stem Cells’ the authors show that polyethylene glycol (PEG) incorporated gold nanoparticles induced adhesion, proliferation and migration of MSCs as well as induction of the SDF-1α/CXCR4 axis associated with MMPs expression. The construct could also prevent MSCs from apoptosis and reactive oxygen species generation. In a small animal model, polyethylene glycol (PEG) incorporated gold nanoparticles enhanced anti-immune response through inhibiting CD86 expression on polarization M1 macrophages upon others.

Major comments:

The manuscript is designed primarily to show effects but does not follow a clear research question and the use of methods to answer the specific aims. Instead, the manuscript sums up individual experiments showing results, some repetitions of previously published observations others with an attempt for novelty. Therefore, the manuscript is not clear enough to reach a major audience with its achievements in the field of MSC research. The authors should provide a clear research question and design the paper accordingly in order to focus on the major tasks.

Major comments:

  • The majority of statements are very superficial and not much evidence is provided to support the statements.
  • The introduction guides the reader from traumatic nerve injury to MSCs. MSCs by secreting growth factors, cytokines, ECM degrading proteases and more, contribute to tissue regeneration. By involving the SDF-1-CXCR4 axis, MSCs capability to migrate can be influenced. Also, the immune-modulatory property of MSCs is cited – all together very superficial – with no direct connection to nerve injury and its regeneration. Along the line, the functional mechanism of MSCs involvement in neuronal recovery is not included at all. Please, rewrite the entire Introduction and include more evidence and information.
  • Nanomaterials demonstrated superior efficiency in drug delivery, tumor therapy and tissue regeneration. This reads like an advertisement. No functional mechanism is provided. Please do not use superior (78), excellent (84), or remarkable (92) in a scientific publication.
  • Appropriate nano biomaterial for tissue regeneration must be biocompatible (102). This way of writing is instructive and not informative because no profound insight follows.
  • The introduction does not follow any clear line. For example, the start of chemistry (102-120) that is needed is initiated too late and therefore the authors start with the different chemical constructs and their interaction with stem cells thereafter but do not include MSCs of reasons I cannot follow. MSCs and their capability were described earlier but their response to chemical or physical cues are absent. This also needs substantial work.
  • The Material and Methods are also superficial. (1) For example the spring constant of the cantilever of the AFM is not provided. (2) No Ethic votum is given that is required when the study uses human material such as placenta and platelets. (3) Where to the platelets come from, how were platelets isolated – by apheresis or were PRPs used? (4) In the process of Western blotting the amount of protein used per lane is important and therefore a quantification is needed with advanced technology. The internal control was beta-actin know to be a second-class house-keeping gene in MSCs since actin is part of the cytoskeleton that changes substantially after chemical and physical manipulation of the MSCs. Also obvious is that the beta-actin is more intense in the lane PEG-Au 43.5 and 174 in Figure 5. (5) Biocompatibility assays comprising cell viability, ROS generation and CD68 expression on monocytes as well as platelet shape analysis is really minimalistic and the same is true for biological examinations. The used methods are very simple out of the catalog and not at all further developed to match with MSCs, monocytes or platelets.
  • In my version of the manuscript the AFM images in Figure 2 are of very poor quality particular the topographic images. The arrows showing filo/lamellipodia in Figure 2F are given with no evidence for correctness – just out of the green. Please give evidence.
  • Figure 3 giving biocompatibility data are also not performed with adequate methods and the same is true for Figure 4, where very poor images are given.
  • Figure 5D shows FACS images with not convincing settings, where every image looks like the control.
  • The data given about the neuronal differentiation are the best of the sets of results provided. Again, the authors used very basic technologies that are not particularly developed for neuronal recovery or repair.
  • The discussion is just a repetition of the results and a real discussion is missing.
  • The conclusion gives ‘PEG bonding with an appropriate amount of Au nanoparticles may produce 791 promising biomaterials which can facilitate neuron tissue regeneration’ with no functional mechanism how the designed scaffold can do so.

Minor comments:

  • The paper needs work on the structural composition of the figures – they are all of poor quality and presented without care for detail.

Author Response

Reviewer 3:

In the paper ‘Inflammatory Modulation of Polyethylene Glycol-AuNP for Regulation of the Neural Differentiation Capacity of Mesenchymal Stem Cells’ the authors show that polyethylene glycol (PEG) incorporated gold nanoparticles induced adhesion, proliferation and migration of MSCs as well as induction of the SDF-1α/CXCR4 axis associated with MMPs expression. The construct could also prevent MSCs from apoptosis and reactive oxygen species generation. In a small animal model, polyethylene glycol (PEG) incorporated gold nanoparticles enhanced anti-immune response through inhibiting CD86 expression on polarization M1 macrophages upon others.

Major comments:

The manuscript is designed primarily to show effects but does not follow a clear research question and the use of methods to answer the specific aims. Instead, the manuscript sums up individual experiments showing results, some repetitions of previously published observations others with an attempt for novelty. Therefore, the manuscript is not clear enough to reach a major audience with its achievements in the field of MSC research. The authors should provide a clear research question and design the paper accordingly in order to focus on the major tasks.

Major comments:

  1. The majority of statements are very superficial and not much evidence is provided to support the statements.

Answer:

We thank the valuable comment from the reviewer. We have carefully modified the manuscript in each section as suggested make it to be easy follow by readers (marked with blue color).

  1. The introduction guides the reader from traumatic nerve injury to MSCs. MSCs by secreting growth factors, cytokines, ECM degrading proteases and more, contribute to tissue regeneration. By involving the SDF-1-CXCR4 axis, MSCs capability to migrate can be influenced. Also, the immune-modulatory property of MSCs is cited – all together very superficial – with no direct connection to nerve injury and its regeneration. Along the line, the functional mechanism of MSCs involvement in neuronal recovery is not included at all. Please, rewrite the entire Introduction and include more evidence and information.

Answer:

We have included more information.

“A literature indicated that SDF-1α/CXCR4 signaling involved in MSCs migration after transplanting MSCs into brain injury of rats. It was demonstrated that SDF-1α was highly expressed in injured area then subsequently stimulate MSCs migration. Furthermore, the amount of BrdU/GFAP positive cells for astrogliosis was also increased in the injured area. The above evidence supported SDF-1α to be a critical factor for cell migration [19].” (Page 2, line 89-95)

  1. Nanomaterials demonstrated superior efficiency in drug delivery, tumor therapy and tissue regeneration. This reads like an advertisement. No functional mechanism is provided. Please do not use superior (78), excellent (84), or remarkable (92) in a scientific publication.

Answer:

We have deleted the wording of “superior, excellent, remarkable” by reviewer’s suggestion.

  1. Appropriate nano biomaterial for tissue regeneration must be biocompatible (102). This way of writing is instructive and not informative because no profound insight follows.

Answer:

We have modified and included more detail description in the “Introduction” section. “Strategies of choosing appropriate nanomaterials and cell model is necessary for tissue regeneration engineering [6]. The poor biocompatibility and lacking of differentiation capacity may lead to failure in clinical treatments [7]. Thus, the mechanical and biological properties of nanomaterials such as biodegradation, cytotoxicity, and ability of anti-immune response should by highly concerned [8].” (Page 2, line 58-62)

  1. The introduction does not follow any clear line. For example, the start of chemistry (102-120) that is needed is initiated too late and therefore the authors start with the different chemical constructs and their interaction with stem cells thereafter but do not include MSCs of reasons I cannot follow. MSCs and their capability were described earlier but their response to chemical or physical cues are absent. This also needs substantial work.

Answer:

Thanks for the suggestion from the Reviewer. We have moved the description in front of “Introduction” section. (Page 2, line 58-79)

  1. The Material and Methods are also superficial. (1) For example the spring constant of the cantilever of the AFM is not provided. (2) No Ethic votum is given that is required when the study uses human material such as placenta and platelets. (3) Where to the platelets come from, how were platelets isolated – by apheresis or were PRPs used? (4) In the process of Western blotting the amount of protein used per lane is important and therefore a quantification is needed with advanced technology. The internal control was beta-actin know to be a second-class house-keeping gene in MSCs since actin is part of the cytoskeleton that changes substantially after chemical and physical manipulation of the MSCs. Also obvious is that the beta-actin is more intense in the lane PEG-Au 43.5 and 174 in Figure 5. (5) Biocompatibility assays comprising cell viability, ROS generation and CD68 expression on monocytes as well as platelet shape analysis is really minimalistic and the same is true for biological examinations. The used methods are very simple out of the catalog and not at all further developed to match with MSCs, monocytes or platelets.

Answer:

  • We have included the more detail description in the “Materials and Methods” section “The range of spring constantfrom around 2.0 N/m.” (Page 5, line 218-219)
  • We have included the IRB approve numbers in the “Materials and Methods” section. (Page 6, line 277)
  • Platelet was obtained and isolated by apheresis from Taichung Veteran Hospital, Taiwan.
  • The quantification data was obtained from three independent experiments for Western Blotting assay. The target proteins (Bcl-2, Bax, Cyclin D1, p21, caspase-3) were prominent expression as well as had significantly expression compare to control group although the internal control of b-actin was slightly increase in the PEG-43.5 and PEG-Au 174 test groups.
  • Biocompatibility assay is an essential process to evaluate the biomaterials property in spite of use common and simple methods. Besides, we also further check and double confirm the biocompatibility and biosafety effect by in vivo assay (Figure 7 and Figure 8) according to the quality in line with this journa

  1. In my version of the manuscript the AFM images in Figure 2 are of very poor quality particular the topographic images. The arrows showing filo/lamellipodia in Figure 2F are given with no evidence for correctness – just out of the green. Please give evidence.

Answer:

(1) We have improved the topographic images of AFM in Figure 2A.

(2) We have included the word mark in the Figure 2E (original Figure 2F).  The following image is the explanation of filo/lamellipodia of cells.

  1. Figure 3 giving biocompatibility data are also not performed with adequate methods and the same is true for Figure 4, where very poor images are given.

Answer:

(1) Biocompatibility assay is an essential process to evaluate the biomaterials property in spite of use common methods as well as published in our previous work. Besides, we also further investigate the biocompatibility and biosafety effect by in vivo assay (Figure 7 and Figure 8) according to the quality in line with this journal.

(2) We have improved the quality of the images in Figure 3 and 4.

  1. Figure 5D shows FACS images with not convincing settings, where every image looks like the control.

Answer:

The quantification data of each group indicates the similar population of viable cells, that demonstrates PEG and PEG-Au (~17.4, ~43.5, ~174 ppm) are good biocompatibility on nanomaterials while culturing with MSCs.

  1. The data given about the neuronal differentiation are the best of the sets of results provided. Again, the authors used very basic technologies that are not particularly developed for neuronal recovery or repair.

Answer:

We will take more time for the execution of all these in vivo experiments to elucidate the neuronal regeneration efficiency. We hope that the reviewer understand the work is very huge and will be too much to be included in the current paper in line with quantify for this journal.

  1. The discussion is just a repetition of the results and a real discussion is missing.

Answer:

We have deleted the repetition and rewritten the Section Discussion.

(1) “Therefore, the treatment with nanomaterials in neural regeneration become an important tissue. However, the materials without good biocompatibility mechanical properties may lead to treatment failure and severe inflammation [45]. Several literatures had pointed out the procedure of autograft implantation had difficulties in neuron repair due to insufficient donor nerves and the donor area may be injured [46], and allografts usually cause failures owing to transplant rejection and serious inflammation [47]. Other biomaterials such as silicone nerve conduit which is a traditional clinical material also has shortcomings due to nonresorbable and decrease in axonal conduction, in addition, it need to be removed via another treatments [48]. Thus, in recent years nanomaterials become a newly issue to improve the neural regeneration engineering.” (Page 20, line 738-748)

(2) “An artificial nanomaterial, carbon nanotube, was considered as a potential material in neuroscience approaches. A literature proved that carbon nanotube scaffolds could affect neuronal signaling, facilitate synaptic plasticity and the formation of neuron networks [49].” (Page 20, line 749-752)

(3) “Nanoparticles such as Au attracted attention in neuroscience field, and it also have been applied in various clinical approaches such as biosensing, drug delivery and tissue regeneration engineering [58]. Due to the physical and chemical properties, Au nanoparticles demonstrated high stability, low cytotoxicity and surface functionalized with polymers [59]. Au nanoparticles can easily interact with cells owing to the small size, that is approximately 1 to 100 nm. Our previous study demonstrated nature molecule such as collagen and fibronectin, fabricated with appropriate amount of Au nanoparticles exhibited superior biocompatibility and induced advanced cell behavior [37, 60]. Moreover, Au nanoparticles have been proved to enhance the growth of neural cells. A previous study indicated chitosan-Au nanoparticles grafted onto poly(D,L-lactide) nerve conduits could induce regeneration in rat sciatic nerve [61]. Another research also demonstrated that near infrared irradiation of Au nanorods can increase the amount of neurons [62]. However, the cetyltrimethylammonium bromide (CTAB) was usually used for the preparation of Au nanorods [63]. According to the research, the chemical residual CTAB after the particle synthesis could induce cytotoxicty and disturb surface hydration of nanoparticles [64]. The physical Au nanoparticles used in current research were collected which as described in a previous study [36], demonstrating low cytotoxicity in vivo and in vitro assessments.” (Page 20-21, line 766-782)

(4) “SDF-1α/CXCR4 pathways and Matrix Metalloproteinase (MMP-2/9) activities regulated stem cell migration and differentiation capacity [65]. A study demonstrated the effect of PEGylated hollow gold nanoparticles (HGNs) could significantly enhance stem cell migration. Furthermore, PEG-HGNs also induced the osteogenic differentiation [66]. Other previous research indicated that PEG can improve the mechanical properties of gelatin hydrogel. The evidence demonstrated gelatin-PEG hydrogel could facilitate the adhesion and proliferation of MSCs and also cell migration [67]. Additionally, a piece of literature has verified that PEG inhibited cell apoptosis via its interactions with mitochondria. These results indicate that PEG improved mitochondrial function and inhibited the release of cytochrome c and pro-apoptotic factors [68]. Previous study also indicated that heparin-functionalized PEG hydrogels supported MSC viability [69].” (Page 21, line 783-793)

(5) “PEG has been proven to increase the expression of neuronal expression markers in the presence of basic fibroblast growth factor (bFGF) and epidermal growth factor (EGF), also enrich the cell population of Neural Precursor Cells (NPCs) [70]. Several studies have also indicated that the PEG-based hydrogel system enhanced adipose tissue regeneration [71] and osteogenesis [72]. In addition, a study also figured out that the properties of culture surface modification could significantly influence the MSCs behavior. Matrigel, the major component are laminin, collagen, entactin and heparan sulfate proteoglycans, at a coating density of 50 μg/cm2 could increase the amount of MSC-derived neuron-like cells with best morphological differentiation and strengthen cell expansion compared to the control group of unmodified polystyrene surface [73]. This indicated an appropriate amount of scaffold substrate can enhance both cell behavior and differentiation capacity. A newly literature has demonstrated that a novel biodegradable hydrogel, oxidized alginate-gelatin-laminin (ADA-GEL-LAM) hydrogel exhibited the ability to promote neuronal differentiation and proliferation of embedded human induced Pluripotent Stem Cells (hiPSCs) [74]. The research went on to add laminin (matrix component) to oxidized alginate-gelatin hydrogels, which effectively facilitated cell adhesion, migration, and differentiation while also suggesting ADA-GEL-LAM as a potential system for neural cell induction.” (Page 21, line 794-811)

(6) “After implanting nanomaterials into injured tissue, the foreign body responses such as inflammation and M2 macrophage polarization were highly concerned [75, 76]. Above all, decorating 43.5 ppm of Au nanoparticles on the PEG polymer significantly enhanced the neurogenic differentiation ability. Although our research team developed a novelty nanomaterial, PEG-Au, with better biocompatibility, biological function, and neuronal differentiation capacity in vitro and in vivo, it still needs more in vivo assessments through the combination with nerve catheter and MSCs to further verify the efficiency in future to provide potential strategies for clinical treatments in neuronal regeneration engineering.” (Page 21, line 815-823)

  1. The conclusion gives ‘PEG bonding with an appropriate amount of Au nanoparticles may produce 791 promising biomaterials which can facilitate neuron tissue regeneration’ with no functional mechanism how the designed scaffold can do so.

Answer:

We have modified “PEG bonding with an appropriate amount of Au nanoparticles may produce promising biomaterials which can facilitate neuron tissue regeneration” to “the findings demonstrated that PEG incorporating with an appropriate amount of Au nanoparticles may be a potential biomaterial for neuronal regeneration engineering”. (Page 22, line 838-839)

Minor comments:

  1. The paper needs work on the structural composition of the figures – they are all of poor quality and presented without care for detail.

Answer:

We have improved the quality of the figures.

Reviewer 4 Report

 In this manuscript, a Polyethylene Glycol (PEG) incorporated with various concentrations of Gold Nanoparticles (Au), was created to investigate the biocompatibility and biological performance in vitro and in vivo.  Numerous results have shown that PEG-Au could be a potential biomaterial, able to cooperate with MSCs for tissue regeneration engineering.

The manuscript shows novelty and it’s interesting for the scientific community.  On the other hand, although it is rich in results, they are presented in an unclear manner and the methods are not fully described. Therefore, manuscript is suitable for the publication in Cells Journal, but major revisions are required as listed below.

Introduction

The introduction is complete and the background is accurate.

On lines 111-112, the English should be correct and the words “(combining what ?)” should be removed.

What about the choice of WJ-SCs as cell model?

The aim of the study should be well explained at the end of the Introduction.

I think that the Figure 9 should become an introductory graphical abstract to help the readers in the comprehension of the different parts of the study: study of the PEG-Au composition, study on cells, study on animals.

Materials and Methods

In the "Methods" section the protocols are not always clearly described and several information is missing. For example:

- there is no indication of the manufacturer for each reagent.

- in the paragraph 2.7 the protocol for the adipogenic differentiation is not described.

In the Paragraph 2.3, specific treatment used for Migration and Differentiation protocols should be removed and moved in the linked paragraphs.

The paragraph 2.4.1 needs to be rewritten. English needs to be improved.

I suggest to check all the paragraphs of the “Materials and Methods” section for English and accuracy of the protocols.

Moreover, approval by ethics committee for the use of WJ-SCs should be specified.

Results

This section should have a subdivision into paragraphs to help the comprehension of the readers. Results are not always clearly described.

Figure 1C: the explanation of the results is not clear and in the Figure it is necessary to change the legend (see PEG-Au).

Figure 2: the description of the panels along the text does not correspond to those shown in the figure. Not all results referring to the figure are described.

Figure 4 E: letters in panel E (a, b, c) should be revised.

Discussion

The experimental design and the data correctly substantiate the results and the discussion, but it is difficult for the reader to indentify in the discussion section the aim and the main results of the study. The interest for the neural differentiation of MSCs does not emerge clearly, becuase too much data is reported. Authors could better describe the limits of this work.

Abbreviations should be explained the first time they appear in the text and then should be used throughout the manuscript.

Check of English grammar and spelling is necessary.

Author Response

Reviewer 4:

In this manuscript, a Polyethylene Glycol (PEG) incorporated with various concentrations of Gold Nanoparticles (Au), was created to investigate the biocompatibility and biological performance in vitro and in vivo.  Numerous results have shown that PEG-Au could be a potential biomaterial, able to cooperate with MSCs for tissue regeneration engineering.

The manuscript shows novelty and it’s interesting for the scientific community.  On the other hand, although it is rich in results, they are presented in an unclear manner and the methods are not fully described. Therefore, manuscript is suitable for the publication in Cells Journal, but major revisions are required as listed below.

Introduction

  1. The introduction is complete and the background is accurate.

Answer:

Thanks for the valuable comment from the Reviewer.

  1. On lines 111-112, the English should be correct and the words “(combining what ?)” should be removed.

Answer:

Thanks for the comment from the reviewer. We have remove and corrected the mistake of “(combining what ?)” (Page 2, line 71)

  1. What about the choice of WJ-MSCs as cell model?

Answer:

The WJ-MSC had potential differentiation capacity not only published in our previous reports but also well widely studied in many literatures due to their superior immune modulation and paracrine secretion ability. Therefore, we have addressed the advantages for the choice of WJ-MSCs as cell model.

  1. The aim of the study should be well explained at the end of the Introduction.

Answer:

Thank the valuable comment from the reviewer. We have included the more detail description in the “Introduction” section “In this study, we have modified functionally AuNP-poly ethylene nanocomposites (PEG-Au) and further exploration the performance of PEG-Au nanocomposites on the immune modulation effect in MSC-based biomaterials application and assessment the potent differentiation capacity of PEG-Au nanocomposites for tissue regeneration.” (Page 3, line 146-149)

  1. I think that the Figure 9 should become an introductory graphical abstract to help the readers in the comprehension of the different parts of the study: study of the PEG-Au composition, study on cells, study on animals.

Answer:

Thanks for the suggestion from the Reviewer. We have changed the Figure 9 into Graphical Abstract.

Materials and Methods

  1. In the "Methods" section the protocols are not always clearly described and several information is missing. For example:

(1) there is no indication of the manufacturer for each reagent.

Answer:

Thanks for the valuable suggestion from the Reviewer. We have included the manufacturer for each reagent.

(2) in the paragraph 2.7 the protocol for the adipogenic differentiation is not described.

Answer:

We have included the protocol for adipogenic differentiation in Section 2.7 “Oil Red O (ORO) staining”. “ORO histochemical analysis was applied to observe adipocyte differentiation. The MSCs at a density of 1´105 were seeded on slides after incubating with various nanomaterials for 7 and 14 days. The MSCs were then fixed by 4% paraformaldehyde for 20 mins and rinsed with 60% isopropanol before proceeding with the staining procedure using ORO (Sigma-Aldrich, 0.35% in isopropanol) and hematoxylin for 10 mins. MSCs were then washed for two times with deionized water and dried at room temperature. The images of stained histology were captured by fluorescence microscopy for further semi-quantification of ORO-positive adipocytes.” (Page 8, line 389-396)

  1. In the Paragraph 2.3, specific treatment used for Migration and Differentiation protocols should be removed and moved in the linked paragraphs.

Answer:

Thanks for the valuable suggestion for the Reviewer. We have move the differentiation protocol to “Materials and Methods” “2.1.3. Reagents of Differentiation Assay For neural differentiation, 1 mM final concentration of β-mercaptoethanol (1 mM, Sigma, USA) was used. For osteogenic differentiation, dexamethasone (0.1 μM, Sigma, USA) and ascorbic acid-2-phosphate (0.25 mM, Sigma, USA) were used. For adipogenic differentiation, dexamethasone (0.1 μM, Sigma, USA), human insulin (0.5 μM, Sigma, USA), and indomethacin (30 μM, Sigma, USA) were used”. (Page 4, line 171-176)

  1. The paragraph 2.4.1 needs to be rewritten. English needs to be improved.

Answer:

We have rewritten the paragraph 2.4.1 “Examination of cell viability “One milliliter of cell suspension containing 2´104 cells (in complete medium with 10% FBS) was injected into each well of the culture plates. For this experiment and all following experiments, cells cultured in a blank well (tissue culture polystyrene, TCPS) were used as control. After incubation, the adherent cells were harvested for 3-(4,5)-dimethylthiahiazo(-z-y1)-3,5-diphenyltetrazolium bromide (MTT) assay. The absorbance was measured at 550 nm with an ELISA reader (F-2500, Hitachi, Japan). Cell morphology was examined by an inverted microscope (TE 300, Nikon, Japan).” (Page 5-6, line 256-262)

  1. I suggest to check all the paragraphs of the “Materials and Methods” section for English and accuracy of the protocols.

Answer:

Thanks for the valuable suggestion from the Reviewer. We have carefully checked the “Materials and Methods” section in the manuscript.

  1. Moreover, approval by ethics committee for the use of WJ-SCs should be specified.

Answer:

Thanks for the suggestion from the Reviewer. We have included the description in Section 2.3 “The MSCs used in the current research were kindly supported by Prof. Woei-Cherng Shyu, which were acquired from the Wharton’s jelly tissue of a human umbilical cord [39].” (Page 5, line 238-239)

Results

  1. This section should have a subdivision into paragraphs to help the comprehension of the readers. Results are not always clearly described.

Answer:

We have made the subdivision into paragraphs by reviewer’s suggestion.

  1. Figure 1C: the explanation of the results is not clear and in the Figure it is necessary to change the legend (see PEG-Au).

Answer:

(1) We have modified the more clearly description in the “Results” section “The FTIR spectra indicated that the specific peaks of pure PEG were 2931 cm-1 (-CH2 vibration), 2868 cm-1 (-CH3) and 1105 cm-1 (OH vibration), and these peaks were also found in PEG-Au 17.4, PEG-Au 43.5, PEG-Au 174 ppm groups. The evidence indicated Au nanoparticles were successfully incorporated with PEG (Figure 1C).” (Page 10, line 465-469); in the “Figure caption” section “The FTIR spectra of pure PEG and PEG-Au (~ 17.4, ~ 43.5, ~ 174 ppm) (Page 10, line 474)

(2) Thanks for the comment from the Reviewer. We have included the legend of Figure 1C.

  1. Figure 2: the description of the panels along the text does not correspond to those shown in the figure. Not all results referring to the figure are described.

Answer:

We have corrected the description in the “Results” section (Page 12, line 513) and “Figure caption” section (Page 11, line 493-501) to make it correspond to Figure 2.

  1. Figure 4 E: letters in panel E (a, b, c) should be revised.

Answer:

We have modified the letters in Figure 4, panel E.

Abbreviations 

  1. Should be explained the first time they appear in the text and then should be used throughout the manuscript.

Answer:

Thanks for the suggestion from the Reviewer. We have checked for the abbreviations in the manuscript.

Check of English 

  1. grammar and spelling is necessary.

Answer:

We have proofread the article to make it well understood by the readers

Round 2

Reviewer 2 Report

Dear authors
since you have answered almost all the questions, I believe that the changes made are sufficient for the publication of the paper. Therefore, in my opinion, the article can be accepted by "cells journal"
Best regards

Author Response

Dear authors
since you have answered almost all the questions, I believe that the changes made are sufficient for the publication of the paper. Therefore, in my opinion, the article can be accepted by "cells journal"
Best regards

Answer: 

Thanks so much the valuable comment. We have appreciated the kindness reply from the reviewer again. 

Reviewer 3 Report

Some crucial issues are still open.

Major comments:

  • The paper ‘Inflammatory Modulation of Polyethylene Glycol-AuNP for Regulation of the Neural Differentiation Capacity of Mesenchymal Stem Cells’ was extensively revised and improved by the authors. Still, the manuscript shows effects but does not follow a clear research question. The manuscript sums up individual experiments, some repetitions of previously published observations others with an attempt for novelty. The authors should provide a clear research question.
  • Material and Methods – cantilever characteristics should be given in detail – AFM.
  • No Ethic votum is given.
  • Beta-actin is a second-class house-keeping gene in MSCs because it reacts to mechanobiological signals and stimuli.

I agree, the manuscript improved but please address these issues in detail.  

Author Response

Comments and Suggestions for Authors

Some crucial issues are still open.

Major comments:

The paper ‘Inflammatory Modulation of Polyethylene Glycol-AuNP for Regulation of the Neural Differentiation Capacity of Mesenchymal Stem Cells’ was extensively revised and improved by the authors. Still, the manuscript shows effects but does not follow a clear research question. The manuscript sums up individual experiments, some repetitions of previously published observations others with an attempt for novelty. The authors should provide a clear research question.

  1. Material and Methods – cantilever characteristics should be given in detail – AFM.

Answer:

We have included the more detail description in the “Materials and Methods” section “(Olympus AC240TS) with low-noise characteristics” (Page 5, line 218)

  1. No Ethic votum is given.

Answer:

We have included the more detail description in the “Materials and Methods” section “IRB approval (CE12164) from Taichung Veteran Hospital”. (Page 6, line 277-278)

  1. Beta-actin is a second-class house-keeping gene in MSCs because it reacts to mechanobiological signals and stimuli.

Answer:

Thanks the valuable comment from the reviewer. b-actin is often used in Western blotting as a loading control, to normalize total protein amounts and check for eventual protein degradation in the samples. Its transcript is also commonly used as a housekeeping gene standard in qPCR. Its molecular weight is approximately 42 kDa.

  1. I agree, the manuscript improved but please address these issues in detail.  

Answer:

Thanks the valuable comment from the reviewer.

Reviewer 4 Report

Dear Authors,

Your manuscript is accurate and full of interesting results.

I Think that You have improved the manuscript as requested.

Therefore, I think the manuscript is now suitable for publication in Cells Journal.

Author Response

Comments and Suggestions for Authors

Dear Authors,

Your manuscript is accurate and full of interesting results. I Think that You have improved the manuscript as requested. Therefore, I think the manuscript is now suitable for publication in Cells Journal.

Answer: 

Thanks so much the valuable comment. We have appreciated the kindness reply from the reviewer again.

Round 3

Reviewer 3 Report

The manuscript is now improved and can be published